# High performance communication by people with paralysis using an intracortical brain-computer interface

Chethan Pandarinath[1,2,3,4,5†], Paul Nuyujukian[1,3,6,7†], Christine H Blabe[1], Brittany L Sorice[8], Jad Saab[9,10,11], Francis R Willett[12,13], Leigh R Hochberg[8,9,10,11,14], Krishna V Shenoy[2,3,6,15,16,17*‡], Jaimie M Henderson[1,3*‡]

[1]Department of Neurosurgery, Stanford University, Stanford, United States; [2]Electrical Engineering, Stanford University, Stanford, United States; [3]Stanford Neurosciences Institute, Stanford University, Stanford, United States; [4]Wallace H. Coulter Department of Biomedical Engineering, Emory University and Georgia Institute of Technology, Atlanta, United States; [5]Department of Neurosurgery, Emory University, Atlanta, United States; [6]Department of Bioengineering, Stanford University, Stanford, United States; [7]School of Medicine, Stanford University, Stanford, United States; [8]Department of Neurology, Massachusetts General Hospital, Boston, United States; [9]School of Engineering, Brown University, Providence, United States; [10]Brown Institute for Brain Science, Brown University, Providence, United States; [11]Center for Neurorestoration and Neurotechnology, Rehabilitation R&D Service, Department of VA Medical Center, Providence, United States; [12]Department of Biomedical Engineering, Case Western Reserve University, Cleveland, United States; [13]Cleveland Functional Electrical Stimulation (FES) Center of Excellence, Louis Stokes VA Medical Center, Cleveland, United States; [14]Department of Neurology, Harvard Medical School, Boston, United States; [15]Neurosciences Program, Stanford University, Stanford, United States; [16]Department of Neurobiology, Stanford University, Stanford, United States; [17]Howard Hughes Medical Institute, Stanford University, Stanford, United States

*For correspondence: shenoy@stanford.edu (KVS); henderj@stanford.edu (JMH)

†These authors also contributed equally to this work
‡These authors also contributed equally to this work

Competing interests: The authors declare that no competing interests exist.

**Abstract** Brain-computer interfaces (BCIs) have the potential to restore communication for people with tetraplegia and anarthria by translating neural activity into control signals for assistive communication devices. While previous pre-clinical and clinical studies have demonstrated promising proofs-of-concept (Serruya et al., 2002; Simeral et al., 2011; Bacher et al., 2015; Nuyujukian et al., 2015; Aflalo et al., 2015; Gilja et al., 2015; Jarosiewicz et al., 2015; Wolpaw et al., 1998; Hwang et al., 2012; Spüler et al., 2012; Leuthardt et al., 2004; Taylor et al., 2002; Schalk et al., 2008; Moran, 2010; Brunner et al., 2011; Wang et al., 2013; Townsend and Platsko, 2016; Vansteensel et al., 2016; Nuyujukian et al., 2016; Carmena et al., 2003; Musallam et al., 2004; Santhanam et al., 2006; Hochberg et al., 2006; Ganguly et al., 2011; O'Doherty et al., 2011; Gilja et al., 2012), the performance of human clinical BCI systems is not yet high enough to support widespread adoption by people with physical limitations of speech. Here we report a high-performance intracortical BCI (iBCI) for communication, which was tested by three clinical trial participants with paralysis. The system leveraged advances in decoder design developed in prior pre-clinical and clinical studies (Gilja et al., 2015; Kao et al., 2016; Gilja et al., 2012). For all three participants, performance exceeded previous iBCIs (Bacher et al., 2015; Jarosiewicz et al., 2015) as measured by typing rate (by a factor of 1.4–4.2) and information throughput (by a factor of 2.2–

4.0). This high level of performance demonstrates the potential utility of iBCIs as powerful assistive communication devices for people with limited motor function.
Clinical Trial No: NCT00912041

## Introduction

Communication is an important aspect of everyday life, achieved through diverse methods such as conversing, writing, and using computer interfaces that increasingly provide an important means to interact with others through channels such as e-mail and text messaging. However, the ability to communicate is often limited by conditions such as stroke, amyotrophic lateral sclerosis (ALS), or other injuries or neurologic disorders which can cause paralysis by damaging the neural pathways that connect the brain to the rest of the body. BCIs offer a potential solution to restore communication by harnessing intact neural signals. Many candidate BCIs have been developed for this purpose, including those based on electroencephalography (*Wolpaw et al., 1998*; *Hwang et al., 2012*; *Spüler et al., 2012*), electrocorticography (*Leuthardt et al., 2004*; *Schalk et al., 2008*; *Moran, 2010*; *Brunner et al., 2011*; *Wang et al., 2013*), and intracortical electrical signals (*Serruya et al., 2002*; *Taylor et al., 2002*; *Carmena et al., 2003*; *Musallam et al., 2004*; *Santhanam et al., 2006*; *Hochberg et al., 2006*; *Ganguly et al., 2011*; *O'Doherty et al., 2011*; *Gilja et al., 2012*; *Simeral et al., 2011*; *Bacher et al., 2015*; *Nuyujukian et al., 2015*; *Aflalo et al., 2015*; *Gilja et al., 2015*; *Jarosiewicz et al., 2015*). Intracortical BCIs (iBCIs), for the purposes of communication in particular, have shown promise in pilot clinical studies (*Bacher et al., 2015*; *Jarosiewicz et al., 2015*). However, iBCIs have not yet reached a level of performance that would support widespread adoption by people with motor impairments that interfere with communication. Further, it is unclear whether current BCI approaches can support high performance during cognitively demanding tasks, such as communicating text.

We recently developed a high-performance iBCI for communication. The BCI provided point-and-click control of a computer cursor (illustrated in *Figure 1a*). Briefly, neural signals (action potentials and high-frequency local field potentials [*Gilja et al., 2012*, *2015*]) were recorded from motor cortex using intracortical microelectrode arrays. These signals were then translated into point-and-click commands using two algorithms developed through prior pre-clinical and clinical research: the ReFIT Kalman Filter for continuous two-dimensional cursor control (*Gilja et al., 2012*, *2015*), and a Hidden Markov Model (HMM)-based state classifier for discrete selection ('click') (*Kao et al., 2016*). To evaluate this interface, we used two approaches: one that represents day-to-day communication use, and one that more rigorously quantifies performance.

## Results

An important real-world application for a communication interface is typing messages in a conversation. We tested whether the BCI could support such an application with T6, a participant in the BrainGate2 pilot clinical trial (http://www.clinicaltrials.gov/ct2/show/NCT00912041). T6 is a 51 year-old woman who was diagnosed with ALS (see Materials and methods: Participants). In these 'free typing' sessions, to simulate use of the BCI in a typical conversation, T6 was prompted with questions and asked to formulate responses at her own pace. Once presented with a question, she was able to think about her answer, move the cursor and click on a button at the bottom right corner of the screen to enable the keyboard, and then type her response (detailed in Materials and methods: Free typing task). T6 typed her responses using an optimized keyboard layout (OPTI-II) (*Rick, 2010*), in which characters are arranged to minimize the travel distance of the cursor while typing English text. T6's mean free typing rate over three days of testing was 24.4 ± 3.3 correct characters per minute (ccpm), which spanned 96 min of typing. (*Figure 1b*; an example free typing video is included as *Video 1*; *Figure 1—figure supplements 1* and *2* list the questions and typed answers from all free typing blocks, and *Figure 1—figure supplement 3* details the filter calibration and assessment stages that preceded the free typing blocks.)

These free typing sessions demonstrated, in a realistic use case, what to our knowledge is the highest typing rate to date by a person with a physical disability using a BCI. However, in the

**eLife digest** People with various forms paralysis not only have difficulties getting around, but also are less able to use many communication technologies including computers. In particular, strokes, neurological injuries, or diseases such as ALS can lead to severe paralysis and make it very difficult to communicate. In rare instances, these disorders can result in a condition called locked-in syndrome, in which the affected person is aware but completely unable to move or speak.

Several researchers are looking to help people with severe paralysis to communicate again, via a system called a brain-computer interface. These devices record activity in the brain either from the surface of the scalp or directly using a sensor that is surgically implanted. Computers then interpret this activity via algorithms to generate signals that can control various tools, including robotic limbs, powered wheelchairs or computer cursors. Such tools would be invaluable for many people with paralysis.

Pandarinath, Nuyujukian et al. set out to study the performance of an implanted brain-computer interface in three people with varying forms of paralysis and focused specifically on a typing task. Each participant used a brain-computer interface known as "BrainGate" to move a cursor on a computer screen displaying the letters of the alphabet. The participants were asked to "point and click" on letters – similar to using a normal computer mouse – to type specific sentences, and their typing rate in words per minute was measured. With recently developed computer algorithms, the participants typed faster using the brain-computer interface than anyone with paralysis has ever managed before. Indeed, the highest performing participant could, on average, type nearly 8 words per minute.

The next steps are to adapt the system so that brain-computer interfaces can control commercial computers, phones and tablets. These devices are widely available, and would allow paralyzed users to take advantage of a range of applications that can be easily downloaded and customized. This development might enable brain-computer interfaces to not only allow people with neurological disorders to communicate, but also assist other people with paralysis in a number of ways.

human-computer interface literature, typing speeds are measured conventionally not in a free typing task, but rather using a 'copy typing' assessment, in which a subject is asked to type pre-determined phrases (reviewed in *MacKenzie and Soukoreff, 2002*). We performed such copy typing assessments with three participants, T6, T5 (a man, 63 years old, with tetraplegia due to spinal cord injury), and T7 (a man, 54 years old, also diagnosed with ALS). Each research session followed a rigorous protocol that aimed to measure peak performance rather than robustness (detailed in Materials and methods: Quantitative performance evaluation and *Figure 2—figure supplements 1* and *2*). Participants were asked to type one of seven sentences (*Figure 2—figure supplement 3*), which were prompted on the screen. Performance was quantified by the number of correct characters typed within each two-minute evaluation block. T6 and T5's performance were assessed using the OPTI-II layout described above as well as a conventional QWERTY layout (*Figure 2a,b*). For participant T7, who had minimal previous typing experience, the QWERTY keyboard was replaced by an alternative layout (ABCDEF; *Figure 2c*), which had the same geometry but with letters arranged in alphabetical order. *Figure 2d* shows examples of prompted and typed text for each participant. We performed five days of testing with T6 (*Figure 2e*; 21 typing evaluation blocks for each keyboard), two days of testing with T5 (*Figure 2f*; 14 typing evaluation blocks for each keyboard), and two days of testing with T7 (*Figure 2g*; 5–6 typing blocks for each keyboard). Example videos that demonstrate cued typing for all participants are included as *Videos 2–7*. T6's average performance using the QWERTY keyboard was $23.9 \pm 6.5$ correct characters per minute (ccpm; mean ± s.d.). T6's average performance using the OPTI-II keyboard was $31.6 \pm 8.7$ ccpm, 1.3 times faster than her performance with the QWERTY layout. Participant T5 averaged $36.1 \pm 0.9$ and $39.2 \pm 1.2$ ccpm for the QWERTY and OPTI-II keyboards, respectively. Participant T7 averaged $13.5 \pm 1.9$ and $12.3 \pm 4.9$ ccpm for the ABCDEF and OPTI-II keyboards, respectively. These results represent a 3.4x (T6, OPTI-II), 4.2x (T5, OPTI-II), and 1.4x (T7, ABCDEF) increase over the previous highest performing BCI report that did not include word completion (9.4 ccpm [*Bacher et al., 2015*]; $p<0.01$ for all three

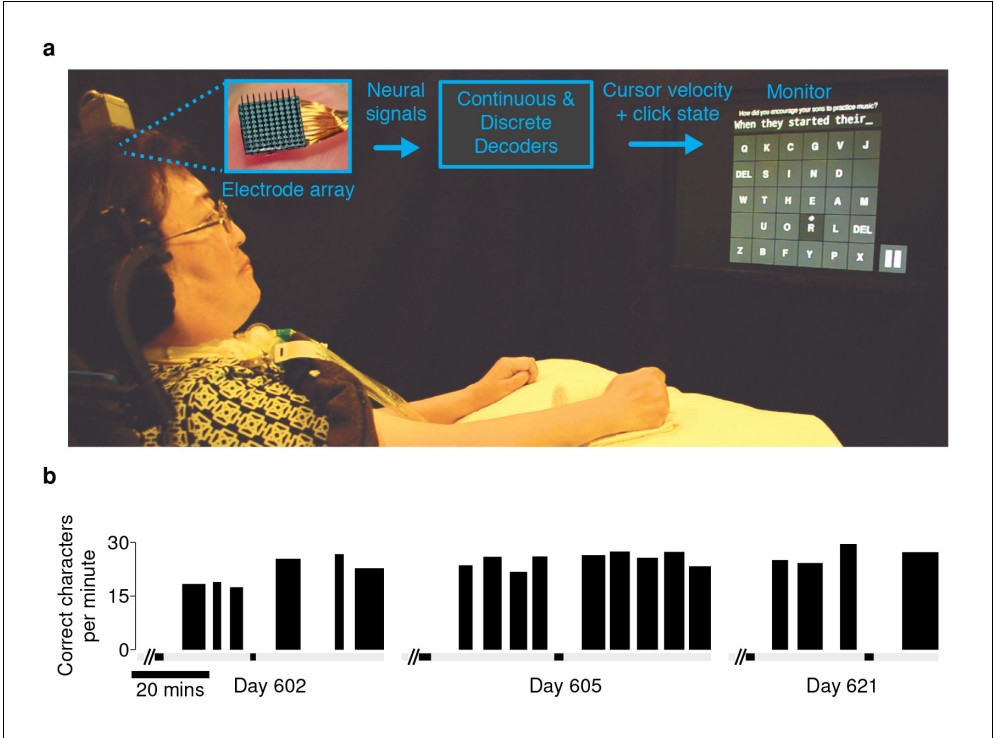

**Figure 1.** Experimental setup and typing rates during free-paced question and answer sessions. (a) Electrical activity was recorded using 96-channel silicon microelectrode arrays implanted in the hand area of motor cortex. Signals were filtered to extract multiunit spiking activity and high frequency field potentials, which were decoded to provide 'point-and-click' control of a computer cursor. (b) Performance achieved by participant T6 over the three days that question and answer sessions were conducted. The width of each black bar represents the duration of that particular block. The black bands along the gray bar just below the black blocks denote filter calibration times. The average typing rate across all blocks was 24.4 ± 3.3 correct characters per minute (mean ± s. d.). *Video 1* shows an example of T6's free typing. The filter calibration and assessment stages that preceded these typing blocks are detailed in *Figure 1—figure supplement 3*.

The following figure supplements are available for figure 1:

**Figure supplement 1.** Participant T6's typed responses during the question and answer sessions.

**Figure supplement 2.** Participant T6's character selection during the question and answer sessions.

**Figure supplement 3.** Filter calibration, assessment, and typing blocks for the 'free typing' sessions performed with participant T6.

participants, single-sided Mann-Whitney *U* tests). Further additions of word completion or prediction should only increase the effective typing rates.

A limitation of the 'copy typing' task is that the performance measurement is affected by the degree of difficulty of each phrase given the specific keyboard being used, as well as the participant's familiarity with the keyboard layouts (e.g., both T5 and T7 had much less familiarity with the keyboard layouts than T6). To explicitly quantify the information throughput of the BCI itself (independent of a phrase or keyboard layout), performance was also measured using a cued-target acquisition task ('grid task' [*Hochberg et al., 2006*; *Nuyujukian et al., 2015*]), in which square targets were arranged in a 6 × 6 grid, and a randomly selected target was cued on each trial. Performance was quantified using 'achieved bitrate' (detailed in *Nuyujukian et al. (2015)* and Materials and methods: Achieved bitrate), which is a conservative measure used to quantify the total amount of information conveyed by the BCI. Briefly, the number of bits transmitted is the net number of correct 'symbols' multiplied by log2(N - 1), where N is the total number of targets. The net

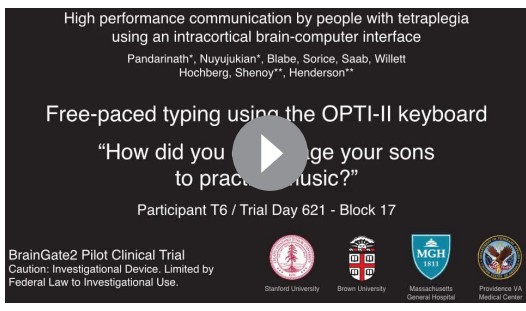

**Video 1.** Example of participant T6's free-paced, free choice typing using the OPTI-II keyboard. T6 was prompted with questions and asked to formulate an answer de novo. Once presented with a question, she was able to think about her answer, move the cursor and click on the play button to enable the keyboard (bottom right corner), and then type her response. In this example, the participant typed 255 characters in ~9 min, at just over 27 correct characters per minute. One of two audible 'beeps' followed a target selection, corresponding to the two possible selection methods: T6 could select targets using either the Hidden Markov Model-based 'click' selection (high-pitched noises) or by 'dwelling' in the target region for 1 s (low-pitched noises). The plot at the bottom of the video tracks the typing performance (correct characters per minute) with respect to time in the block. Performance was smoothed using a 30 s symmetric Hamming window. The scrolling yellow bar indicates the current time of that frame. During the free typing task, T6 was asked to suppress her hand movements as best as possible. (During the quantitative performance evaluations, T6 was free to make movements as she wished.) This video is from participant T6, Day 621, Block 17. Additional 'free typing' examples for T6 are detailed in *Figure 1—figure supplements 1* and *2*.

(i.e., comparing T6's free typing vs. copy typing) only accounted for a ~30% performance difference, rather than the 2–4x performance difference between studies. Thus, cognitive load is unlikely to account for the differences in performance.) The performance increase over previous work is unlikely to be due to experience with BCI, as participants in the current study had a similar range of experience using the BCI as those in comparable studies (*Table 2*). Example videos that demonstrate the grid task for all participants are included as *Videos 8–11*. In addition, comparisons of the HMM's performance against the previous highest-performing approach for discrete selection are presented in *Figure 3—figure supplement 1*. We performed additional grid measurements with T5 in which targets were arranged in a denser grid (9 × 9). This task allows

number of correct symbols is taken as the total number of correct selections minus the total number of incorrect selections, i.e., each incorrect selection requires an additional correct selection to compensate (analogous to having to select a keyboard's backspace key). For example, on an eight-target task, if the net rate of correct target selections (after compensating for incorrect selections) were one target / s, the achieved bitrate would be 2.8 bits / s.

Over 5 days of testing with T6 (*Figure 3a*; 21 grid evaluation blocks), 4 days of testing with T5 (*Figure 3b*; 29 grid evaluation blocks) and 2 days of testing with T7 (*Figure 3c*; six grid evaluation blocks), average performance was 2.2 ± 0.4 bits per second (bps; mean ± s.d.), 3.7 ± 0.4 bps, and 1.4 ± 0.1 bps, respectively. This is a substantial increase over the previous highest achieved bitrates for people with motor impairment using a BCI (*Table 1*), which were achieved by two of the same participants in an earlier BrainGate study (T6: 0.93 bits / s, T7: 0.64 bits / s, from *Jarosiewicz et al. (2015)*; p<0.01 for both participants, single-sided Mann-Whitney *U* tests). For T6 and T7, who participated in the previous study, performance of the current methods represents a factor of 2.4 (T6) and 2.2 (T7) increase. For T5, the current performance represents a factor of 4.0 increase over the highest performing participant in the previous study. (The previous study measured performance using a free typing task, which includes the cognitive load of word formation [*Jarosiewicz et al., 2015*]. However, the effects of cognitive load in the current study

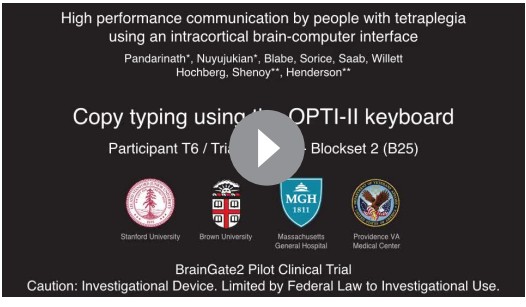

**Video 2.** Example of participant T6's 'copy typing' using the OPTI-II keyboard. In the copy typing task, participants were presented with a phrase and asked to type as many characters as possible within a two-minute block. T6 preferred that the cursor remain under her control throughout the task – i.e., no re-centering of the cursor occurred after a selection. This video is from participant T6, Day 588, Blockset 2. Performance in this block was 40.4 ccpm.

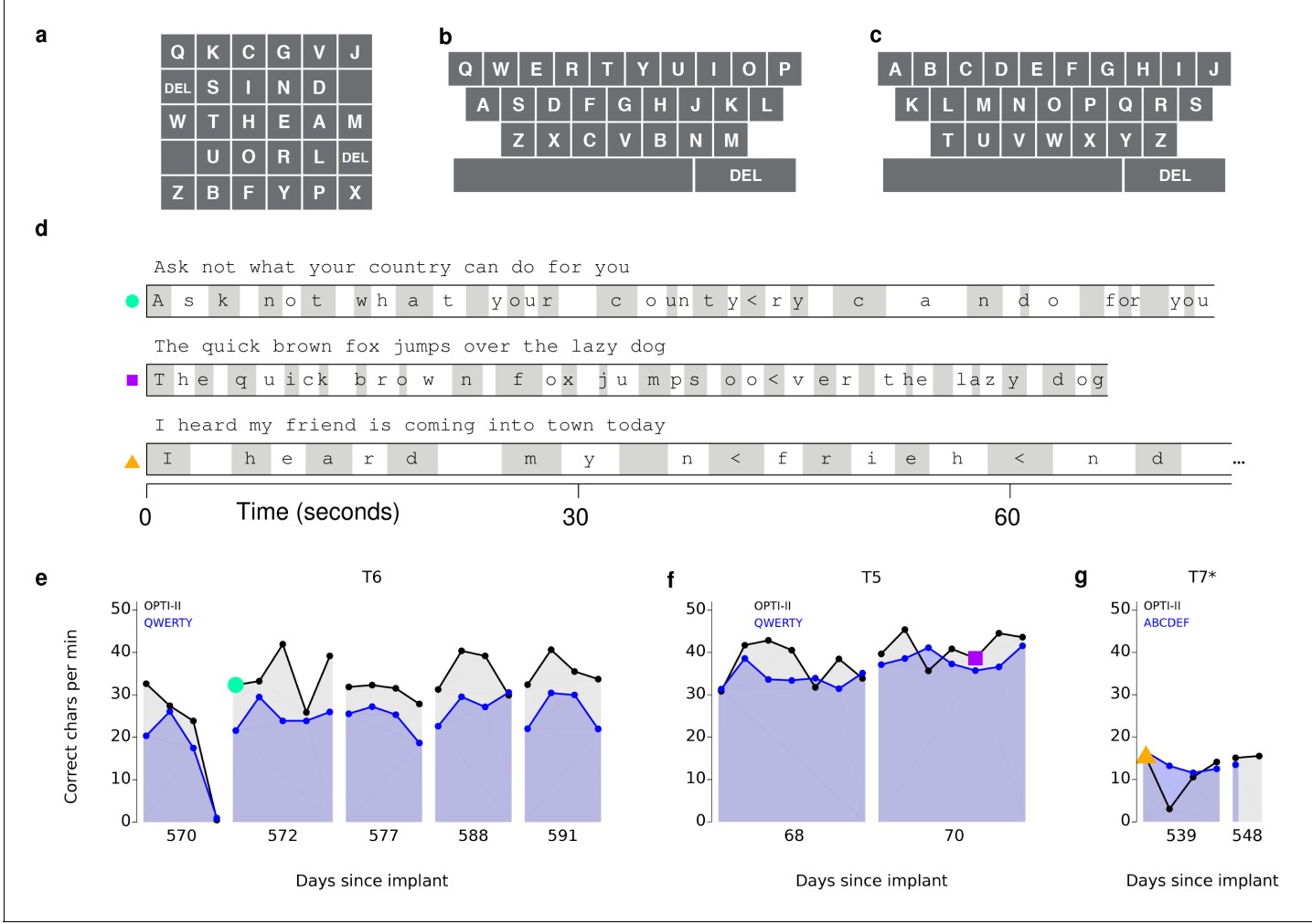

**Figure 2.** Performance in copy typing tasks. (**a**) Layout for the OPTI-II keyboard (**b**) Layout for the QWERTY keyboard. (**c**) Layout for the ABDEF keyboard. (**d**) Examples of text typed during three copy typing evaluations with participants T6, T5, and T7. Each example shows the prompted text, followed by the characters typed within the first minute of the two-minute evaluation block. Box width surrounding each character denotes the time it took to select the character. '<' character denotes selection of a backspace key. Colored symbols on the left correspond to blocks denoted in lower plots. (**e**) Performance in the copy typing task with the QWERTY (blue) and OPTI-II (black) keyboards across 5 days for participant T6. QWERTY performance was 23.9 ± 6.5 correct characters per minute (ccpm; mean ± s.d.), while OPTI-II performance was 31.6 ± 8.7 ccpm. X-axis denotes number of days since array was implanted. (**f**) Performance in the copy typing task with the QWERTY (blue) and OPTI-II (black) keyboards across 2 days for participant T5. Average performance was 36.1 ± 0.9 and 39.2 ± 1.2 ccpm for the QWERTY and OPTI-II keyboards, respectively. (**g**) Performance in the copy typing task with the ABCDEF (blue) and OPTI-II (black) keyboards across two days for participant T7. Average performance was 13.5 ± 1.9 and 12.3 ± 4.9 ccpm for the ABCDEF and OPTI-II keyboards, respectively. *Participant T7 did not use an HMM for selection.

The following figure supplements are available for figure 2:

**Figure supplement 1.** Data collection protocol for quantitative performance evaluation sessions.

**Figure supplement 2.** Example of the blockset structure for quantitative performance evaluation sessions.

**Figure supplement 3.** Sentences used to evaluate performance in copy typing tasks.

**Table 1.** Survey of BCI studies that measure typing rates (correct characters per minute; ccpm), bitrates, or information transfer rates for people with motor impairment. Number ranges represent performance measurements across all participants for a given study. Communication rates could be further increased by external algorithms such as word prediction or completion. As there are many such algorithms, the current work excluded word prediction or completion to focus on measuring the performance of the underlying system. The most appropriate points of comparison, when available, are bitrates, which are independent of word prediction or completion algorithms. Similarly, information transfer rates are also a meaningful point of comparison, though they are less reflective of practical communication rates than bitrate (which takes into account the need to correct errors; detailed in **Nuyujukian et al. (2015)**; **Townsend et al. (2010)**). For the current work, and for Jarosiewicz et al. 2015, we also break down performance by individual participant to facilitate direct comparisons (denoted by italics). As shown, performance in the current study outperforms all previous BCIs tested with people with motor impairment. *These numbers represent performance when measured using a denser grid (9 × 9; **Figure 3—figure supplement 2** and **Video 10**). **For this study, reported typing rates included word prediction / completion algorithms. ***Number range represents the range of performance reported for the single study participant. ****Other reported numbers included word prediction / completion algorithms. †Acronyms used: ReFIT-KF: Recalibrated Feedback Intention-trained Kalman Filter. HMM: Hidden Markov Model. CLC: Closed-loop Calibration. LDA: Linear Discriminant Analysis. RTI: Retrospective Target Inference. DS: Dynamic Stopping.

| Study | Participant (s) | Recording modality | Control modality | Etiology of motor impairment | Average typing rate (ccpm) | Average bitrate (bps) | Average ITR (bps) |
|---|---|---|---|---|---|---|---|
| This study | average (N = 3) | intracortical | ReFIT-KF +HMM† | ALS (2), SCI (1) | 28.1 | 2.4 | 2.4 |
| '' | T6 | | | ALS | 31.6 | 2.2 | 2.2 |
| '' | T5 | | | SCI | 39.2 | 3.7 | 3.7 |
| '' | '' | | | '' | - | 4.2* | 4.2* |
| '' | T7 | | (No HMM) | ALS | 13.5 | 1.4 | 1.4 |
| Bacher et al., 2015 | S3 | intracortical | CLC+LDA† | brainstem stroke | 9.4 | - | - |
| Jarosiewicz et al., 2015 | average (N = 4) | intracortical | RTI+LDA† | ALS (2), brainstem stroke (2) | n/a** | 0.59 | - |
| '' | T6 | | | ALS | '' | 0.93 | - |
| '' | T7 | | | ALS | '' | 0.64 | - |
| '' | S3 | | | brainstem stroke | '' | 0.58 | - |
| '' | T2 | | | brainstem stroke | '' | 0.19 | - |
| Nijboer et al., 2008 | N = 4 | EEG | P300 | ALS | 1.5–4.1 | - | 0.08–0.32 |
| Townsend et al., 2010 | N = 3 | EEG | P300 | ALS | - | 0.05–0.22 | - |
| Münßinger et al., 2010 | N = 3 | EEG | P300 | ALS | - | - | 0.02–0.12 |
| Mugler, et al. 2010 | N = 3 | EEG | P300 | ALS | - | - | 0.07–0.08 |
| Pires et al., 2011 | N = 4 | EEG | P300 | ALS (2), cerebral palsy (2) | - | - | 0.24–0.32 |
| Pires et al., 2012 | N = 14 | EEG | P300 | ALS (7), cerebral palsy (5), Duchenne muscular dystrophy (1), spinal cord injury (1) | - | - | 0.05–0.43 |
| Sellers et al., 2014 | N = 1 | EEG | P300 | brainstem stroke | 0.31–0.93*** | - | - |
| McCane et al., 2015 | N = 14 | EEG | P300 | ALS | - | - | 0.19 |
| Mainsah et al., 2015 | N = 10 | EEG | P300-DS† | ALS | - | - | 0.01–0.60 |
| Vansteensel et al., 2016 | N = 1 | subdural ECoG | Linear Classifier | ALS | 1.15**** | - | 0.21 |

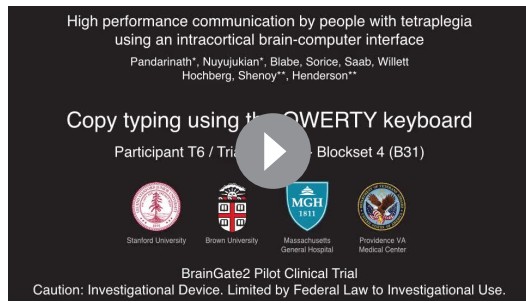

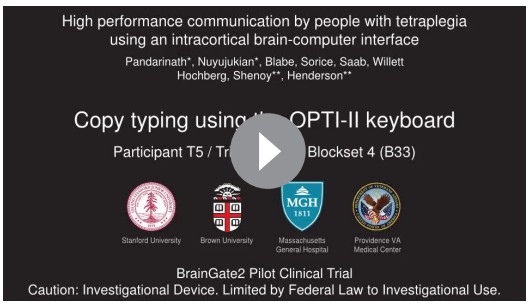

**Video 3.** Example of participant T6's 'copy typing' using the QWERTY keyboard. Same as *Video 2*, but using the QWERTY keyboard layout. This video is from participant T6, Day 588, Blockset 4. Performance in this block was 30.6 ccpm.

**Video 4.** Example of participant T5's 'copy typing' using the OPTI-II keyboard. Same as *Video 2*, but for participant T5. This video is from participant T5, Day 68, Blockset 4. Performance in this block was 40.5 ccpm.

the possibility for higher bitrates than the 6 × 6 grid used above, with the tradeoff that selecting these smaller targets requires higher control fidelity. Across two days of testing with T5 (*Figure 3—figure supplement 2* and *Video 10*; 8 evaluation blocks), average performance was 4.16 ± 0.39 bps, which was significantly greater than the 6 × 6 performance (p<0.01, Student's *t* test) and represents, to our knowledge, the highest documented BCI communication rate for a person with motor impairment.

We note that in both sets of quantitative performance evaluations (copy typing and grid tasks), participant T6, who retained significant finger movement abilities, continued to move her hand while controlling the BCI. Further research sessions, in which T6 was asked to suppress her natural movements to the best of her abilities, showed similar performance in both copy typing and grid tasks (detailed in *Figure 4* and supplements, which quantify her performance and the degree to which she was able to suppress movements). As might be expected, T6 found that suppressing her natural movement was a challenging, cognitively demanding task. Though she did this to the best of her abilities, the act of imagining finger movement still elicited minute movements, both during 'open-loop' decoder calibration where she was imagining movements, and during closed-loop control of the BCI. While we were unable to record EMG activity (as permission to do so had not previously been sought), we were able to record the movements of her fingers using a commercially-available 'dataglove' sensor system. This was also used for research sessions in which decoder calibration was based on her physical movements. Overall, when T6 actively attempted to suppress movements, her movement was reduced by a factor of 7.2–12.6 (*Figure 4—figure supplement 1*). Despite this factor of 7.2–12.6 in movement suppression, performance was quite similar to performance when T6 moved freely - across all three quantitative evaluation types (Grid, OPTI-II, QWERTY), the performance differences were within 0–20% and not significant (p>0.2 in all cases, Student's *t* test).

## Discussion

The high-performance BCI demonstrated here has potential utility as an assistive communication system. The average copy typing rates demonstrated in this study were 31.6 ccpm (6.3 words per minute; wpm), 39.2 ccpm (7.8 wpm), and 13.5 ccpm (2.7 wpm) for T6, T5, and T7, respectively. In surveying people with ALS, (*Huggins et al., 2011*) found that 59% of respondents would be satisfied with a

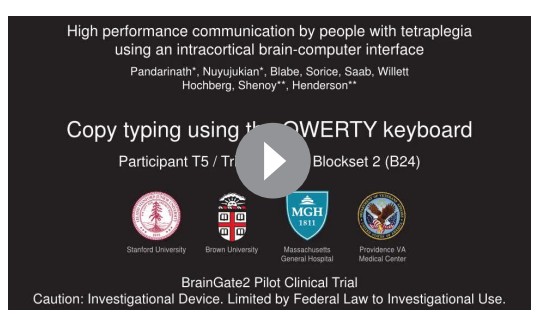

**Video 5.** Example of participant T5's 'copy typing' using the QWERTY keyboard. Same as *Video 4*, but using the QWERTY keyboard layout. This video is from participant T5, Day 68, Blockset 2. Performance in this block was 38.6 ccpm.

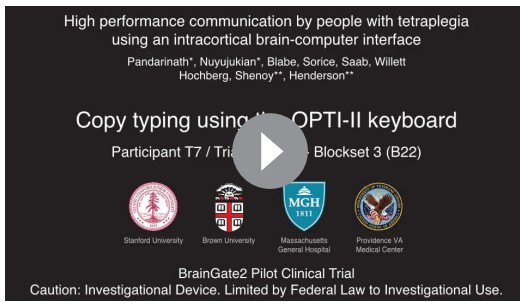

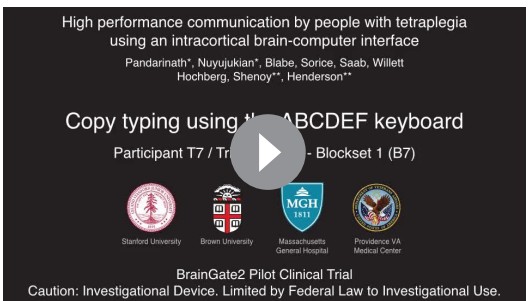

**Video 6.** Example of participant T7's 'copy typing' using the OPTI-II keyboard. Same as *Video 2*, but for participant T7. T7 selected letters by dwelling on targets only. In addition, T7 preferred that the cursor re-center after every selection (i.e., following a correct or an incorrect selection). These across-participant differences are detailed in Materials and methods: Quantitative performance evaluations (under 'Target selection and cursor re-centering'). This video is from participant T7, Day 539, Blockset 3. Performance in this block was 10.6 ccpm.

**Video 7.** Example of participant T7's 'copy typing' using the ABCDEF keyboard. Same as *Video 6*, but using the ABCDEF keyboard layout. This video is from participant T7, Day 539, Blockset 1. Performance in this block was 16.5 ccpm.

communication BCI that achieved 10–14 ccpm (2–2.8 wpm), while 72% would be satisfied with 15–19 ccpm (3–3.8 wpm). Thus, the current performance would likely be viewed positively by many people with ALS. Current performance still falls short of typical communication rates for able-bodied subjects using smartphones (12–19 wpm [*Hoggan et al., 2008*; *Lopez et al., 2009*]), touch typing (40–60 wpm [*MacKenzie and Soukoreff, 2002*]), and speaking (90–170 spoken wpm [*Venkata-giri, 1999*]); continued research is directed toward restoring communication toward rates that match able-bodied subjects.

Previous clinical studies of intracortical BCIs have either used generalized (task-independent) measures of performance (*Simeral et al., 2011*; *Gilja et al., 2015*) or application-focused (task-dependent) measures (*Bacher et al., 2015*; *Jarosiewicz et al., 2015*; *Hochberg et al., 2012*; *Collinger et al., 2013a*). While application-focused measurements are crucial in demonstrating clinical utility, performance might be heavily dependent on the specific tasks used for assessment. By rigorously quantifying both generalized performance (grid task) and application-specific performance (copy typing task) with all three participants, we aim to provide helpful benchmarks for continued improvement in neural decoding and BCI communication interface comparisons.

Another critical factor for demonstrating clinical utility is characterizing the day-to-day variability often seen in BCI performance. To do so we approached the quantitative performance evaluation sessions (grid and copy typing) with a strict measurement protocol (similar to *Simeral et al., 2011*), and did not deviate from this protocol once the session had begun. Inclusion of detailed measurement protocols will help in demonstrating the repeatability (or variability) of various BCI approaches and establish further confidence as BCIs move closer to becoming more broadly available for people who would benefit from assistive communication technologies. The grid task and bit rate assessment described previously and in this manuscript may serve as a valuable task and metric to document further progress in BCI decoding.

As mentioned earlier, our quantitative performance evaluation protocol was designed to measure peak performance in a repeatable manner rather than measuring the system's stability. To standardize the performance measurements, explicit decoder recalibration or bias re-estimation blocks were performed prior to each measurement set (as detailed in Materials and methods: Quantitative performance evaluation and *Figure 2—figure supplements 1* and *2*). A key additional challenge for clinically useful BCIs is maintaining system stability, and future work will combine our performance-driven approach with complementary approaches that focus on achieving long-term stability without explicit recalibration tasks (*Jarosiewicz et al., 2015*).

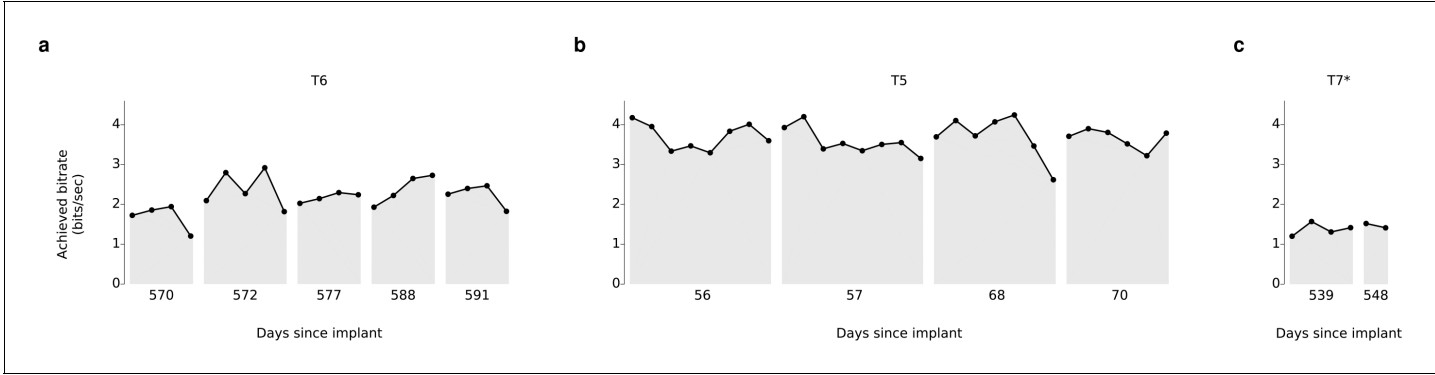

**Figure 3.** Information throughput in the grid task. (a) Performance in the grid task across 5 days for participant T6. T6 averaged 2.2 ± 0.4 bits per second (mean ± s.d.). (b) Performance in the grid task across 2 days for participant T5. T5 averaged 3.7 ± 0.4 bits per second. (c) Performance in the grid task across 2 days for participant T7. T7 averaged 1.4 ± 0.1 bits per second. X-axis denotes number of days since array was implanted. *Participant T7 did not use an HMM for selection.

The following figure supplements are available for figure 3:

**Figure supplement 1.** Performance of the HMM-based classifier during grid tasks with participants T6 and T5.

**Figure supplement 2.** Information throughput for participant T5 when using a dense grid.

The typing rates achieved in this study were performed without any word completion or prediction algorithms. While such algorithms are commonly used in input systems for mobile devices and assistive technology, our aim in this report was to explicitly characterize the performance of the intracortical BCI, without confounding the measurement by the choice of a specific word completion algorithm (of which there are many). Important next steps would be to apply the BCI developed here to a generalized computing interface that includes word completion and prediction algorithms to further boost the effective communication rates of the overall system. Regardless of the assistive platform chosen, all systems would benefit from higher performing BCI algorithms. We also note that the data for participants T6 and T7 was collected 1.5 years after neurosurgical placement of the intracortical recording arrays. This, along with other recent reports (*Gilja et al., 2012*; *Simeral et al., 2011*; *Nuyujukian et al., 2015*; *Gilja et al., 2015*; *Hochberg et al., 2012*; *Chestek et al., 2011*; *Bishop et al., 2014*; *Flint et al., 2013*; *Nuyujukian et al., 2014*), demonstrates that intracortical BCIs may be useful for years post-implantation.

Central to the results demonstrated with participants T6 and T5 was the identification of independent control modalities to simultaneously support high performance continuous control and discrete

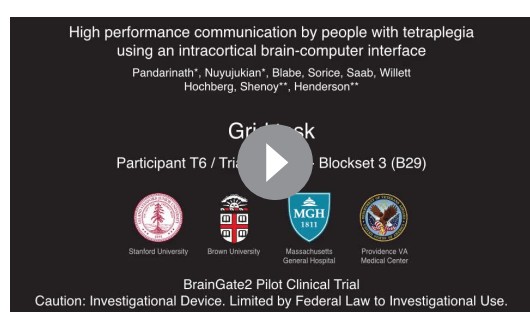

**Video 8.** Example of participant T6's performance in the grid task. This video is from participant T6, Day 588, Blockset 3. Performance in this block was 2.65 bps.

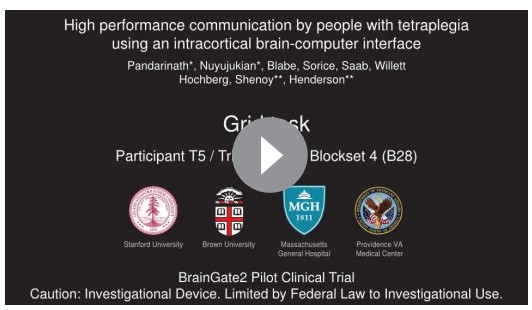

**Video 9.** Example of participant T5's performance in the grid task. This video is from participant T5, Day 56, Blockset 4 (Block 28). Performance in this block was 4.01 bps.

**Table 2.** Participants' prior BCI experience and training for studies considered in **Table 1**. The experience column details the number of participants in the respective study that had prior experience with BCIs at the time of the study and, if reported, the duration of that prior experience and/or training.

| Study | Participant(s) | BCI experience/training |
| --- | --- | --- |
| This study | average (N = 3) | 1 year |
| '' | T6 | 1.5 years |
| '' | T5 | 9 prior sessions (≈1 month) |
| '' | T7 | 1.5 years |
| *Bacher et al., 2015* | S3 | 4.3 years |
| *Jarosiewicz et al., 2015* | average (N=4) | *2 years* |
| '' | T6 | *10 months to 2.3 years* |
| '' | T7 | *5.5 months to 1.2 years* |
| '' | S3 | *5.2 years* |
| '' | T2 | *4.6 months* |
| *Nijboer et al., 2008* | N = 4 | At least 4–10 months |
| *Townsend et al., 2010* | N = 3 | All had prior P300 BCIs at home, two had at least 2.5 years with BCIs |
| *Münßinger et al., 2010* | N = 3 | Two of three had prior experience, training not reported |
| *Mugler, et al. 2010* | N = 3 | Average experience of 3.33 years |
| *Pires et al., 2011* | N = 4 | No prior experience, training not reported |
| *Pires et al., 2012* | N = 14 | Not reported |
| *Sellers et al., 2014* | N = 1 | Prior experience not reported, thirteen months of continuous evaluation |
| *McCane et al., 2015* | N = 14 | Not reported |
| *Mainsah et al., 2015* | N = 10 | Prior experience not reported, two weeks to two months of evaluation |
| *Vansteensel et al., 2016* | N = 1 | 7 to 9 months |

selection. Specifically, we found that activity on T6's array had the highest neural modulation when attempting or imagining movements of her contralateral thumb and index finger, and further, that these two independent effectors could be merged to provide closed-loop control of a single effector (cursor). We also found that this thumb and index finger-based control modality increased system robustness and yielded decoders that were more resilient to nonstationarities. Finally, we found that a separate behavioral approach, ipsilateral hand squeeze, provided an independent, readily-

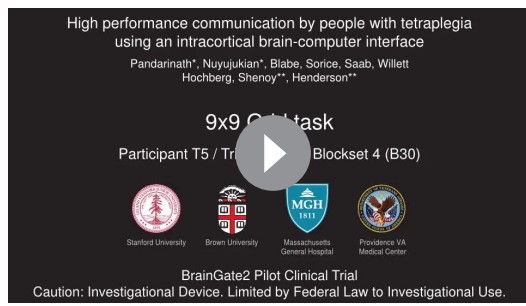

**Video 10.** Example of participant T5's performance in the dense grid task (9 × 9). This video is from participant T5, Day 56, Blockset 4 (Block 30). Performance in this block was 4.36 bps.

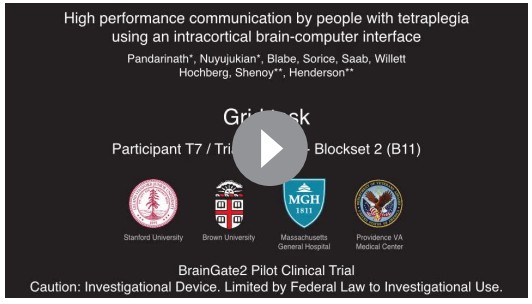

**Video 11.** Example of participant T7's performance in the grid task. This video is from participant T7, Day 539, Blockset 2. Performance in this block was 1.57 bps.

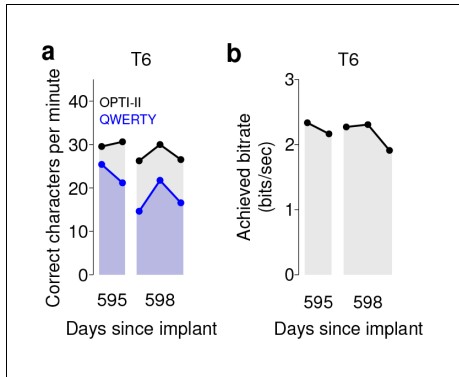

**Figure 4.** Performance of the BCI with movements suppressed. A potential concern is that the demonstrated performance improvement for participant T6 relative to previous studies is due to her retained movement ability. Participant T6 was capable of dexterous finger movements (as opposed to participants T5 and T7, who retained no functional movements of their limbs). To control for the possibility that physical movements underlie the demonstrated improvement in neural control, we measured T6's BCI performance during the same quantitative performance evaluation tasks, but asked her to suppress her movements as best as she could. In these sessions, decoders were calibrated based on imagined (rather than attempted) finger movements. (a) During copy typing evaluations with movements suppressed, T6's average performance using the OPTI-II keyboard was 28.6 ± 2.0 ccpm (mean ± s.d.), and her average performance using the QWERTY keyboard was 19.9 ± 4.3 ccpm (as discussed in the main text, her performance while moving freely was 31.6 ± 8.7 ccpm and 23.9 ± 6.5 ccpm for the OPTI-II and QWERTY keyboards, respectively). (b) During grid evaluations with movements suppressed, T6's achieved bitrate was 2.2 ± 0.17 bps (compared to 2.2 ± 0.4 bps while moving freely). We note that using the BCI while suppressing movements is a more difficult and cognitively demanding task - since the participant's natural, intuitive attempts to move actually generate physical movements, she needed instead to imagine movements, and restrict her motor cortical activity to patterns that do not generate movement. (This is supported by the participants own comment regarding the difficulty in controlling the BCI while imagining movement without actually moving: 'It is a learning curve for me to not move while imagining.') Despite this additional cognitive demand, performance with movements suppressed was quite similar to performance when the participant moved freely (within 0–20%) - in all three cases, the differences in performance were not significant (p>0.2 in all cases, Student's *t* test). Data are from T6's trial days 595 and 598.

The following figure supplement is available for figure 4:

*Figure 4 continued on next page*

combined control dimension to support discrete selection. We performed a similar protocol for evaluating behavioral imagery strategies with participant T5 and found his highest neural modulation was elicited when imagining movements of the whole arm. We combined this imagery strategy with ipsilateral hand squeeze (mirroring findings from participant T6) to yield simultaneous high performance continuous control and discrete selection.

The BCI approach demonstrated here was first developed with participant T6 and then adapted for participant T7. However, initially, we often found that instabilities would appear in T7's control on shorter timescales (i.e., across tens of minutes). In these instances, biases in the cursor's velocity would develop that impeded high performance control. To counteract these effects, we introduced a variant of the bias correction method used in *Jarosiewicz et al. (2015)*; *Hochberg et al. (2012)* with T7 (detailed in Materials and methods), which continuously estimated and corrected biases during closed-loop BCI use and resulted in more stable control. Further, instead of calibrating a new decoder between measurement sets (as was done for T6), we found it was sufficient to keep the decoder constant and simply perform a short target acquisition task to estimate and update the underlying bias estimate. We therefore incorporated this revised protocol (holding decoders constant, and simply updating the underlying bias estimate) for sessions with T5.

The performance achieved by all participants in this study outperformed all previous BCIs for communication tested with people with motor impairment. However, we note that T6 and T5's communication rates were substantially better than those of T7. Many factors could have contributed to this difference in performance. Certainly, with any skilled motor task, one expects to see variation in performance across participants, even in able-bodied subjects (e.g. playing sports or musical instruments). As ALS is a disease with a large degree of variance in its effects, participant-specific differences in disease effects or progression may play a role in the differences in performance between T6 and T7. Interestingly, we note that in the center-out-and-back task (where reaction times can be most easily measured), T7 demonstrated increased response latency relative to T6. Specifically, the time between the appearance of a cued target and neural modulation corresponding to a movement attempt was more than 100 ms later for T7 relative to T6. It is unclear whether this additional

*Figure 4 continued*

**Figure supplement 1.** Participant T6's movements are greatly reduced when movements are actively suppressed.

latency was due to variability in the effects of ALS across participants. Differences in the participants' prior experiences may have also played a role: T6 was much more familiar with computing devices, while T7 rarely used them. This difference in familiarity / comfort with text entry may have contributed to the difference in typing rates.

At the time of this study, participant T6 still retained the ability to make dexterous movements of her hands and fingers, which may raise the question of whether her high level of performance was related to the generation of movement. As described in the Results section, to test the effects of movement generation on BCI control, we performed separate sessions in which T6 suppressed her movement to the best of her ability, and found no measurable effect on BCI performance. This result is consistent with previous studies that have evaluated the effects of movement on BCI control. For example, Gilja*, Nuyujukian* et al. (*Gilja et al., 2012*) compared BCI performance in non-human primates while their arms were either restrained or able to move freely, and found little difference in performance. Additionally, Ethier et al (*Ethier et al., 2012*). showed that monkeys whose grasping movements were prevented using a paralytic agent were still able to reliably generate grasping-related cortical activity, which could then be decoded to activate a functional electrical stimulation system that restored grasping ability. Multiple participants with no movement of their limbs have also successfully controlled a computer cursor or other external device through this intracortical BCI (e.g., ref. [*Hochberg et al., 2006*], participants S1 and S2, and ref. [*Hochberg et al., 2012*], participant T2). Finally, we recently investigated the effects of movement on cursor control quality in detail with clinical trial participants and found no decrease in performance when movements were suppressed (*Gilja et al., 2015*).

In this study we have controlled for the potential issue of movement generation as closely as is possible given the proper boundaries of clinical research. We have presented data from three participants, two of whom had no ability to make functional arm or hand movements, and one who suppressed her movements to the best of her abilities, below a range in which the movements could be functionally useful. All three cases are representative examples of arm and hand movement capabilities of the severely motor impaired population, and, in all three cases, the participants communicated with the BCI at rates that exceeded any previous study of people with motor impairment. Further, there was little if any correspondence between the participants' movement abilities and BCI performance.

Both participants T6 and T5 used the HMM decoder for discrete selection. Our goal was to also use the HMM with participant T7. However, he passed away (from causes unrelated to the trial) before we were able to perform those research sessions. As mentioned above, we initially found that neural features with participant T7 exhibited drifts in baseline firing rates over time, which necessitated the integration of strategies to mitigate the effects of these baseline drifts on continuous cursor control. Thus, our plan for data collection was to first develop these strategies and carefully document performance with T7 using continuous cursor control only, and subsequently add the HMM for discrete selection. The first part was successful – as shown, T7 achieved high quality continuous control, and the resultant communication performance was double that of the previous highest-performing approach. Unfortunately, however, T7 passed away before the HMM sessions could be conducted.

Previous work with non-human primates from our lab and others (*Musallam et al., 2004*; *Santhanam et al., 2006*; *Shenoy et al., 2003*) demonstrated that BCI strategies which leverage discrete classification can achieve high communication rates. The 'point-and-click' approach demonstrated in the current paper (i.e., continuous control over a cursor's movement, plus a decoder for discrete selection [*Simeral et al., 2011*; *Bacher et al., 2015*; *Jarosiewicz et al., 2015*]) was investigated instead because it has certain practical advantages over the classification approach. In particular, developing a robust point-and-click controller provides a flexible interface that can be applied to a wide variety of computing devices. A point-and-click controller could be integrated with mobile computing interfaces (i.e., smartphones or tablets) that would dramatically increase what is achievable with the BCI, without the need for the development of custom software for each function (as would be needed for a discrete interface). Finally, and perhaps most fundamentally, as this approach

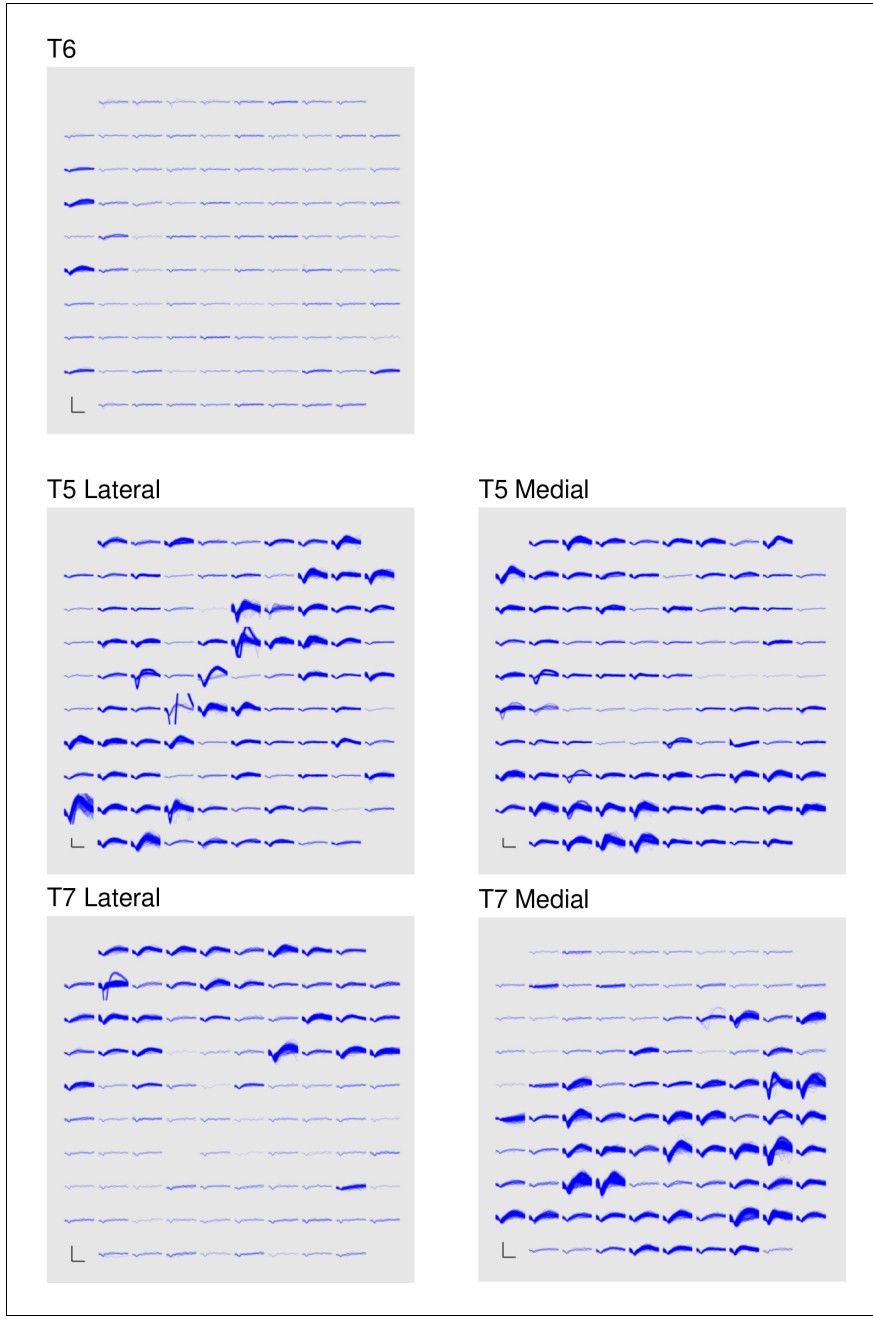

**Figure 5.** Signal quality on the participants' electrode arrays. Each panel shows the recorded threshold crossing waveforms for all 96 channels of a given array for a 60 s period during the participant's first quantitative performance evaluation block. T6 had a single implanted array, while T5 and T7 had two implanted arrays. Scale bars (lower left corner of each panel) represent 150 µV (vertical) and 0.5 milliseconds (horizontal). Voltages were analog band-pass filtered between 0.3 Hz and 7.5 kHz, then sampled by the NeuroPort system at 30 kHz. The resulting signals were then digitally high-pass filtered (500 Hz cutoff frequency) and re-referenced using common average referencing. Thresholds were set at −4.5 times the root-mean-squared (r.m.s.) voltage value for each channel. Channels without a corresponding trace did not have any threshold crossing events during this time period. Data are from sessions 570, 56, and 539 days post-implant for T6, T5, and T7, respectively.

The following figure supplements are available for figure 5:

**Figure supplement 1.** HF-LFP signals have similar time course and condition dependence to spiking activity.

*Figure 5 continued*

**Figure supplement 2.** HF-LFP signals show a similar time course and condition dependence to spiking activity during auditory-cued tasks in which the participant had her eyes closed.

enables both continuous movements and selections, it is more general as long as performance is high (as reported here). Thus, point-and-click interfaces are a key step to creating BCIs that allow flexible, general-purpose computing use.

The discrete classification approach provides a promising alternative strategy for communication BCIs. However, there are multiple technical challenges to the previously demonstrated approaches, and multiple unknowns when translating these approaches to people. From a technical standpoint, one of the primary challenges is that a multi-class discrete classifier may need a specified time window over which to classify neural features into a discrete selection. In the earlier high-performance study with non-human primates (*Santhanam et al., 2006*), this necessitated a 'fixed pace' design in which the monkeys were prompted to make a sequence of selections at a fixed timing interval. Such an approach may prove more difficult with people typing messages, which requires the flexibility to actively think about what to type and to type at a free pace. A potential approach to enable a 'free-paced' BCI was demonstrated in follow-on studies (*Achtman et al., 2007*; *Kemere et al., 2008*), which showed in offline analyses that state transitions could be inferred automatically from neural activity, thus automatically detecting the necessary time window for classification. But this has not been demonstrated in closed-loop experiments by our group or, to our knowledge, by other groups. Thus there are multiple technical and scientific challenges to address, and developing these approaches for clinical trial participants is an active area of research.

The collaborative approach reported here involved carrying out the same investigative protocol by independent teams at sites across the country. This approach supports replication with multiple participants to go beyond initial proof of principle, but presented its own challenges, particularly in designing and implementing closed-loop BCI approaches remotely. Specifically, iterating on decoder designs and troubleshooting performance issues greatly benefits from real-time access to system performance and data. To this end, the development of a framework for remote, real-time performance monitoring (detailed in Materials and methods: System design) was critical to understanding and iteratively addressing performance issues during research sessions with remote participants.

The participants' comments provided insight on the subjective experience of using the BCI. All three participants commented on the ease-of-use of the system. Participant T6 compared the BCI system to other assistive communication devices, remarking: 'The one I like is this one as opposed to an eye gaze system... It is quite intuitive.' Similarly, participant T7 noted: 'When things go well, it feels good.' – this statement was a comment on the improvement in control after weeks of development and testing (using the approach outlined above). Additionally, participant T5 compared his

**Table 3.** Summary of the decoding and calibration approaches used with each participant.

|  | T6 | T7 | T5 |
|---|---|---|---|
| Continuous decoding algorithm | ReFIT Kalman Filter (threshold crossings and HF-LFP) | ReFIT Kalman Filter (threshold crossings) | ReFIT Kalman Filter (threshold crossings) |
| Discrete decoding algorithm | Hidden Markov Model (HF-LFP) | n/a | Hidden Markov Model (threshold crossings) |
| Dwell time | 1 s (reset on target exit) | 1.5 s (cumulative) | 1 s (reset on target exit) |
| Bias estimation | no | yes | yes |
| Cursor recentering | no | yes | no |
| Recalibration blocks | Recalibrated continuous and discrete decoders | Only updated bias estimates | Only updated bias estimates |
| Error attenuation in recalibration block | yes | yes | no |

typing performance to his standard typing interface (a head mouse-based tracking system), noting 'After I typed using BrainGate for 2 days, the weekend came and I went back to my existing typing system and it was ponderously slow.' Interestingly, participant T6 also noted that the motor imagery used for filter calibration did not match the imagery she found most effective during closed-loop BCI control. Specifically, while T6's continuous control was calibrated based on index finger and thumb movement imagery, T6 commented that during closed-loop BCI control, 'It feels like my right hand has become a joystick.'

The question of the suitability of implanted versus external BCI systems (or any other external AAC system) for restoring function is an important one. Any technology (or any medical procedure) that requires surgery will be accompanied by some risk; among the most immediate risks that should be considered with any neurosurgery involving a craniotomy include bleeding, infection, seizure, and headache. That risk is not viewed in isolation, but is compared – by the individual contemplating the procedure – to the potential benefit (*Hochberg and Cochrane, 2013*; *Hochberg and Anderson, 2012*). There are several important factors one might take into consideration, for example ease-of-use, cosmesis, and performance. Any externally applied BCI system (EEG for example) will require donning and doffing, meaning that it could not be used continuously 24 hr a day. A future self-calibrating, fully implanted wireless system could in principle be used without caregiver assistance, would have no cosmetic impact, and could be used around the clock. Such a system may be achievable by combining the advances in this report with previous advances in self-calibration and in fully-implantable wireless interfaces (*Jarosiewicz et al., 2015*; *Borton et al., 2013*). Additional discussion of these topics are found in refs. (*Ryu and Shenoy, 2009*; *Gilja et al., 2011*).

In a recent survey of people with spinal cord injury (*Blabe et al., 2015*), respondents with high cervical spinal cord injury would be more likely to adopt a hypothetical wireless intracortical system compared to an EEG cap with wires, by a margin of 52% to 39%. In another survey, over 50% of people with spinal cord injury would 'definitely' or 'very likely' undergo an implant surgery for a BCI (*Collinger et al., 2013b*). Thus, there is a clear willingness among people with paralysis to undergo a surgical procedure if it could provide significant improvements in their daily functioning.

In summary, we demonstrated a BCI that achieved high performance communication in both free typing and copy typing, leveraging system design and algorithmic innovations demonstrated in prior pre-clinical and clinical studies (*Gilja et al., 2012*, *2015*; *Kao et al., 2016*). Using this interface, all three participants achieved the highest BCI communication rates for people with movement impairment reported to date. These results suggest that intracortical BCIs offer a promising approach to assistive communication systems for people with paralysis.

## Materials and methods

Permission for these studies was granted by the US Food and Drug Administration (Investigational Device Exemption) and Institutional Review Boards of Stanford University (protocol # 20804), Partners Healthcare/Massachusetts General Hospital (2011P001036), Providence VA Medical Center (2011–009), and Brown University (0809992560). The three participants in this study, T6, T7, and T5, were enrolled in a pilot clinical trial of the BrainGate Neural Interface System (http://www.clinicaltrials.gov/ct2/show/NCT00912041). Informed consent, including consent to publish, was obtained from the participants prior to their enrollment in the study. Additional permission was obtained to publish participant photos and reproduce text typed by the participants.

### Participants

Participant T6 is a right-handed woman, 51 years old at the start of this work, who was diagnosed with Amyotrophic Lateral Sclerosis (ALS) and had resultant motor impairment (functional rating scale (ALSFRS-R) measurement of 16). In Dec. 2012, a 96-channel intracortical silicon microelectrode array (1.0 mm electrode length, Blackrock Microsystems, Salt Lake City, UT) was implanted in the hand area of dominant motor cortex as previously described (*Simeral et al., 2011*; *Hochberg et al., 2012*). T6 retained dexterous movements of the fingers and wrist. Data reported in this study are from T6's post-implant days 570, 572, 577, 588, 591, 602, 605, and 621.

A second study participant, T7, was a right-handed man, 54 years old at the time of this work, who was diagnosed with ALS and had resultant motor impairment (ALSFRS-R of 17). In July 2013, participant T7 had two 96-channel intracortical silicon microelectrode arrays (1.5 mm electrode

length, Blackrock Microsystems, Salt Lake City, UT) implanted in the hand area of dominant motor cortex. T7 retained very limited and inconsistent finger movements. Data reported are from T7's post-implant days 539 and 548. Unfortunately, prior to performing additional research sessions, T7 passed away due to non-research related reasons.

A third study participant, T5, is a right-handed man, 63 years old at the time of this work, with a C4 ASIA C spinal cord injury that occurred approximately 9 years prior to study enrollment. He retains the ability to weakly flex his left (non-dominant) elbow and fingers; these are his only reproducible movements of his extremities. He also retains some slight residual movement which is inconsistently present in both the upper and lower extremities, mainly seen at ankle dorsiflexion and plantarflexion, wrist, fingers and elbow, more consistently present on the left than on the right. Occasionally, the initial slight voluntary movement triggers involuntary spastic flexion of the limb. In Aug. 2016, participant T5 had two 96-channel intracortical silicon microelectrode arrays (1.5 mm electrode length, Blackrock Microsystems, Salt Lake City, UT) implanted in the upper extremity area of dominant motor cortex. During BCI control, the only observed movement of the extremities (besides involuntary spastic flexion) is finger flexion on the non-dominant hand during discrete selection attempts. Data reported are from T5's post-implant days 56, 57, 68, and 70.

## System design

Data were collected using the BrainGate2 Neural Interface System. This modular platform, standardized across clinical trial sites, supports multiple operating systems and custom real-time software, and allows multiple studies to be performed by different researchers without hardware modification. The framework enables a rapid-prototyping environment and facilitates ease of replication of real-time closed loop studies with multiple trial participants. For the present study, neural control and task cuing closely followed ref. (*Gilja et al., 2015*) and were controlled by custom software running on the Simulink/xPC real-time platform (The Mathworks, Natick, MA), enabling millisecond-timing precision for all computations. Neural data were collected by the NeuroPort System (Blackrock Microsystems, Salt Lake City, UT) and available to the real-time system with 5 ms latency. Visual presentation was provided by a computer via a custom low latency network software interface to Psychophysics Toolbox for Matlab and an LCD monitor with a refresh rate of 120 Hz. Frame updates from the real-time system occurred on screen with a latency of approximately $13 \pm 5$ ms.

During design and development research sessions leading up to the quantitative performance evaluations, a framework for remote, real-time performance monitoring and debugging was critical to iteratively improving system performance remotely. This was performed using lightweight, custom MATLAB software that monitored performance via network packets from the real-time system and provided insight to researchers located in the laboratory (i.e., away from the participants' homes, where data were collected). Researchers accessed the remote monitoring and troubleshooting system in real-time using TeamViewer (Tampa, FL). In addition, at the end of each evaluation block, summary data was immediately transferred from the participants' homes to researchers to facilitate rapid analysis, debugging, and iteration.

## Neural feature extraction

The neural signal processing framework closely followed ref. (*Gilja et al., 2015*). The NeuroPort System applies an analog 0.3 Hz to 7.5 kHz band-pass filter to each neural channel and samples each channel at 30 kSamples per second. These broadband samples were processed via software on the Simulink/xPC real-time platform. The first step in this processing pipeline was to subtract a common average reference (CAR) from each channel (intended to remove noise common to all recorded neural channels). For each time point, the CAR was calculated simply by taking the mean across all neural channels.

Band-pass filters split the signal into spike and high frequency local field potential (HF-LFP) bands. To extract neural spiking activity, a cascaded infinite impulse response (IIR) and finite impulse response (FIR) high-pass filter were applied. A threshold detector was then applied every millisecond to detect the presence of a putative neural spike. Choice of threshold was specific to each array (T6: $-50$ μV, T5, $-95$ μV, Medial and Lateral arrays; T7, Lateral array: $-70$ μV, Medial array: $-90$ μV). HF-LFP power features refer to the power within the 150–450 Hz band-pass filtered signal. For continuous control, T6 sessions used both spike and HF-LFP features (hybrid decoding), while T5 and T7

sessions used only spike-based features. *Figure 5* demonstrates the signal quality for both participants.

A potential concern with decoding a power signal such as these high frequency LFP (HF-LFP) signals (which were used for participant T6) is that they may pick up artifacts related to EMG from eye movements. In intracranial studies, such artifacts have been previously shown in electrocortographic (ECoG) recordings (e.g., Kovach et al [*Kovach et al., 2011*].). However, as demonstrated in Kovach et al., the magnitude of this phenomenon falls sharply with the distance from the ventral temporal cortical surface. Further, the same study demonstrated that these artifacts are highly correlated across scales less than 1 cm, and that rereferencing on these local scales eliminates the artifacts outside of the immediate ventral temporal cortical surface (Kovach et al., Fig. 9). In our study, data are collected in motor cortical areas which are fairly medial in the precentral gyrus (frontal lobe), and are rereferenced using the common average across the intracortical array (4 mm x 4 mm). Given the large distance between the recording site and the ventral temporal cortical surface, and the common average rereferencing, any minor eye movement-related EMG artifacts are expected to be essentially eliminated.

In order to be certain that these artifacts do not play a role, we provide additional lines of evidence that rule out EMG due to eye movements as being the driver of the observed high performance. First, we include data from an additional participant (T5) in which HF-LFP signals were not used for control (*Figures 2 and 3*). We found T5's performance was greater than T6's – this demonstrates that high performance iBCI control is achievable using spiking activity alone. Second, we analyzed T6's HF-LFP signals during decoder calibration blocks and show that they have a similar time course and condition dependence as recorded spiking activity (*Figure 5—figure supplement 1*). Third, we include additional data recorded as T6 performed an auditory-cued task with her eyes closed as she attempted movements of her fingers, wrist, and elbow (*Figure 5—figure supplement 2*). Because there are no visual cues and the participant has her eyes closed, it is unlikely that the participant is making condition-dependent eye movements. However, even in the absence of visual cues, the HF-LFP signals are quite similar to recorded spiking activity in their time course and condition dependence. These lines of evidence make the possibility that HF-LFP signals are eye movement-related highly unlikely.

During sessions with participant T7, neural features exhibited drifts in baseline firing rates over time. To account for these nonstationarities, baseline rates were computed de novo prior to each block, during a 30 s period in which the participant was asked simply to relax.

## Neural control algorithms

Two-dimensional continuous control of the cursor used the ReFIT Kalman Filter (detailed in refs. [*Gilja et al., 2012*, *2015*]). For participants T6 and T5, discrete selection ('click') was achieved using a Hidden Markov Model (HMM)-based state classifier, which was previously developed with non-human primates (*Kao et al., 2016*) and adapted for the current work. At each timestep, the HMM calculated the probability that the participant's intended state was either movement or click. For T6, only HF-LFP features were used in the HMM, while only spike features were used for T5. Features were pre-processed with a dimensionality reduction step using Principal Components Analysis (PCA). The HMM classified the probability of state $s_k$ as:

$$p(s_{k,t}) = p(s_k|z_t) \sum_i p(s_{k,t}|s_{i,t-1}) p(s_{i,t-1}),$$

where $p(s_k|z_t)$ is the probability of being in state $s_k$ given the current (dimensionality-reduced) neural features at time $t$, $z_t$, and where $p(s_{k,t}|s_{i,t-1})$ denotes the probability of transitioning from state $s_i$ to state $s_k$. $p(s_k|z_t)$ was modeled as a multivariate Gaussian distribution with separate mean and covariance for each state. The current state was classified as 'click' when $p(s_k, t)$ exceeded a pre-determined threshold that was calculated in an unsupervised fashion (threshold choice is outlined in the task descriptions).

In this framework, there is a key tradeoff between including more PCs (and potentially more relevant information) and overfitting / mis-estimating the mean and covariance of the Gaussian distribution for each state as more dimensions (PCs) are added. Overfitting these parameters results in poor decoding on 'out-of-sample' data. Empirically, we found that 3–4 PCs resulted in an HMM that

accurately classified states without overfitting on the limited training data. Therefore, the top four eigenvalue-ranked PCs were kept and used as inputs to the HMM.

Algorithm parameters were calibrated using training data collected during the same research session as evaluation of neural control performance. All calibration data were collected with a center-out-and-back target configuration. For the quantitative evaluations, initial filters were calibrated based on data collected during a center-out-and-back task performed under motor control (T6) or automated open-loop control (all T5 and T7). During motor control tasks, T6 controlled the cursor's x and y velocities using index finger and thumb movements, respectively, and acquired targets by holding the cursor still over the target (dwell tasks) or squeezing her left hand (ipsilateral to the implanted array) when the cursor was over the target (click tasks). T6's physical movements were recorded using left- and right-handed datagloves (5DT, Irvine, CA). This was not performed for participants T7 and T5 because they did not have functional use of their arms or hands. During automated open-loop calibration (T5 and T7), the cursor's movements followed pre-programmed trajectories, and the participants attempted movements to follow the cursor's movement. In addition, during open-loop calibration, T5 attempted to squeeze his left hand to acquire targets. After initial filter calibration, both continuous control and discrete filters were then recalibrated using closed-loop neural control data. This closed-loop recalibration block closely followed (*Gilja et al., 2015*), with the addition of a discrete selection for T6 and T5. Because the quality of the initial VKF filter varied from day to day, the recalibration blocks for T6 and T7 also used error attenuation (*Hochberg et al., 2012*; *Velliste et al., 2008*) to ensure that the participant could reach all targets.

For participant T6, to control for the possibility that her ability to generate movements led to her high performance, we performed additional sessions in which she was asked to suppress her movements as best as she could (outlined in *Figure 4* and supplements). For these sessions, to calibrate the initial continuous and discrete filters, T6 performed an automated open-loop filter calibration protocol as described above. This protocol was also followed for the free typing evaluations (outlined in *Figure 1* and supplements).

Neural features used in each filter were selected during the filter calibration process. For the ReFIT-KF, features were first ranked by tuning significance (i.e., p-value of the linear regression between binned neural data and cursor velocity). Features were then added one by one in order of tuning significance to the neural control algorithm, and an offline assessment of directional control was used to predict online control quality. The set of features chosen was the one that minimized the number of features used while maximizing cross-validated decoding accuracy. The discrete decoder (HMM) used all available HF-LFP features.

For both the continuous cursor-positioning ReFIT-KF decoder and the discrete click-state HMM decoder, neural data were binned every 15 ms and sent through the decoders. Thus, for the ReFIT-KF decoder, updated cursor velocity estimates were provided every 15 ms for use in the rest of the BMI system. This velocity was integrated to update the cursor position estimate every 1 ms, and therefore the most recent cursor position was sent to the display every 1 ms. The computer monitor was updated every 8.3 ms (i.e., at the 120 Hz frame rate of the monitor) with the most recent estimate of the desired cursor position. The high update rate is important so as to not inadvertently and deleteriously add latency into the BMI which is a closed-loop feedback control system (*Cunningham et al., 2011*) and which was possible by using a commercially available high-speed monitor. This system design and these timings are consistent with our previous work (Gilja*, Nuyujukian* et al. Nat Neurosci 2012 [*Gilja et al., 2012*] binned neural data and used the ReFIT-KF to decode every 50 ms; Gilja*, Pandarinath*, et al. Nat Med 2015 [*Gilja et al., 2015*] binned neural data and used the ReFIT-KF to decode every 10–50 ms depending on the experiment; a 120 frame/s monitor was also employed). This operates faster and more accurately than a recent report claims is possible with a Kalman filter (*Shanechi et al., 2017*), and at a higher level of performance than recently reported (*Shanechi et al., 2017*).

As the HMM click decoder facilitates a discrete decision, a threshold criteria for selection was needed. This threshold value was set after each retraining block at the 93rd quantile of state estimates for the respective retraining block. When running in closed loop, after two consecutive 15 ms bins where the HMM click state probability was above this threshold value, the system generated a click and selected the target under the cursor.

As mentioned above, during sessions with participant T7, neural features exhibited drifts in baseline firing rates over time. On short timescales, these drifts manifested as biases in decoded

velocities. Biases were reduced using a variant of the bias correction method used in *Jarosiewicz et al. (2015)*; *Hochberg et al. (2012)*, with the addition of a magnitude term that corrects for the frequency of observed speeds (i.e., low speeds are generally observed for longer time periods than high speeds). Specifically, e.g. for the $x$ direction, velocity bias was estimated as:

$$B_x(t) = B_x(t-1) + (V_x(t) - B_x(t-1)) \times |V_x(t) - B_x(t-1)| \times \Delta t / \tau,$$

where $B_x(t)$ represents the bias estimate for the $x$ direction at time $t$, $V_x(t)$ represents the velocity estimate for the $x$ direction at time $t$, $\Delta$t is the time step of adaptation (0.001 s), and $\tau$ controls the adaptation rate (we set $\tau$ to 30 s). A larger $\tau$ makes the system slower to respond to changes in bias, but reduces the size of transient fluctuations in the bias estimate when no actual bias is present. The current bias estimate was only updated when speed exceeded a threshold (threshold was set to be roughly the 10–20% quantile for the speeds typically observed for T7).

Data from participant T5 was collected after data collection from participants T6 and T7, and the scientific protocol used with participant T5 reflected the advances made with the prior two participants. *Table 3* highlights these changes. As participant T5 and T7's arrays had a large number of highly modulated spiking channels, no HFLP was necessary to build their decoders. After participant T6's data collection was completed, it was discovered that both cursor movement and click decoders can be calibrated during the initial open loop block, and this approach was used with participant T5. Similarly, the bias correction algorithm was implemented for participant T7 after data collection for participant T6 was completed and it was discovered that the cursor movement decoder did not need to be retrained after every blockset (a bias correction update block would suffice). This time-saving approach was also used with participant T5.

## Free typing task

The aim of this task was to create a natural, familiar, and conversational environment to demonstrate the potential for iBCIs to be used as communication devices. In this task, conducted only with participant T6, questions were presented at the top of the monitor. These questions were tailored to topics that T6 enjoys discussing. At the start of a block, one of these questions would be chosen from a pool of questions that had not been used before. After reading the question considering her response, T6 started the block counter and enabled the keyboard inputs by selecting the play button in the bottom right corner of the screen. She then used the BCI to type her response to the question by selecting one letter at a time.

During the free typing task, T6 was asked to suppress her hand movements as best as possible. During the quantitative performance evaluations, T6 was free to make movements as she wished.

## Quantitative performance evaluations

The quantitative measurement experiments were performed with all three participants. These experimental days were explicitly structured and carefully timed so that each piece of data could be compared and measured independently. The experimental flow diagram for participant T6 is shown in *Figure 2—figure supplement 1*. With participants T6 and T5, the calibration protocol resulted in two BCI decoders: one for cursor movement and one for click. With participant T7, only a cursor movement decoder was calibrated. After decoders were calibrated and confirmed to be working successfully in a brief (less than 30 s evaluation), the experimental data were then collected. Once the data portion of the experiment was started, the blockset structure was repeated until the participant ended the research session. Starting over with the calibration portion of the protocol was not permitted once the blockset data collection portion of the research day began.

### Blocksets

Each blockset was collected in a strict, timed, randomized fashion. Each blockset was considered a complete and independent unit, equally weighted, and statistically identical to all other blocksets. Blockset timing structure is defined in *Figure 2—figure supplement 2*. Each blockset began with a recalibration block, which resulted in new cursor movement and click decoders for participant T6. For participants T5 and T7, the movement decoder was held constant and the recalibration block was simply used to create an updated estimate of the underlying velocity bias. This recalibration protocol was used to maximize the performance of the data collected in the time-locked blocks that followed. Three data blocks were then collected in a randomized fashion, constituting a blockset. Each

blockset consisted of one block each of three tasks. The three tasks with participant T6 and T5 were the grid task, the QWERTY task, and the OPTI-II task. With participant T7, the QWERTY task was substituted with the ABCDEF task, since he had minimal experience with the conventional (QWERTY) keyboard layout. The task order in each blockset was randomized subject to the constraint that the two copy typing tasks were always adjacent. This constraint minimized the amount of elapsed time between the copy typing blocks, in order to minimize any confounding effects on measured typing rate. The prompted sentence to copy in both keyboard tasks for a given blockset was identical. Following the completion of a blockset, participants were given as long a break (to request a drink from a caregiver, etc.) as desired before starting the subsequent blockset. Breaks within a blockset were minimized as best as possible.

### Target selection and cursor re-centering

For both participants, selections could be made by holding the cursor over the target for a fixed period of time (1 s for T6 and T5, 1.5 s for T7). For T6 and T5, leaving a given target area would reset the hold time counter to 0 – thus they were required to remain over the same target for a full second to select via holding. For T7, who could only select targets by dwelling on them, selection used a strategy called 'cumulative dwell time' – each target had a separate hold time counter, and the cumulative time spent over a target counted towards the 1.5 s requirement (i.e., it was not required that the 1.5 s be contiguous). All hold time counters were reset to 0 after any target selection. Additionally, T6 and T5 could also select targets using the HMM-based click decoder, which was typically a faster method of selecting targets. Thus, T6 and T5 had two methods for target selection.

Participant T7 preferred that the cursor re-center to the middle of the screen after each selection, which allowed him to better focus on one trajectory at a time. (In contrast, participants T6 and T5 preferred continuous cursor control instead of re-centering, as it allowed them to plan out a series of keystrokes and achieve faster typing rates.)

For T7, after each selection, the cursor was centered relative to the targets and held in place for 500 ms – during this time, target selection was disabled. This approach minimized the 'worst case' path lengths (i.e., eliminated the potential of having to move from one corner of the keyboard to another while typing a phrase); this is beneficial in the case where instability causes control biases, which make long trajectories that oppose the bias more difficult. We note that, as re-centering was completely unsupervised (i.e., it occurred regardless of whether the selection made was correct), it did not compromise the typing or achieved bitrate measurements in any way.

### Grid task

The purpose of this task was to measure performance using information theoretic metrics. In this grid task (*Hochberg et al., 2006*; *Nuyujukian et al., 2015*), the workspace was divided into a 6 × 6 grid of equal gray squares. Each square was selectable, and one would randomly be prompted as the target when illuminated in green. After a selection was made, a new target was immediately prompted. This task ran for two minutes (fixed duration).

### QWERTY task

The purpose of this task was to measure typing rates using a conventional keyboard layout. In this task, a sentence was prompted at the top of the screen, and participants were instructed to copy this sentence as quickly and accurately as possible. Selection methods were identical to that described in the grid task. This task ended when the participants typed the last letter of the prompted sentence or two minutes had elapsed, whichever occurred first.

### ABCDEF task

The purpose of this task was identical to the QWERTY task, except it was specific to participant T7. Since he was not very familiar with the QWERTY layout, the letters were rearranged alphabetically from left to right, top to bottom. This alphabetical ordering allowed T7 to more easily determine where a given letter was located. The keyboard geometry of the ABCDEF task was identical to that of the QWERTY task, and the same task timing and prompting was employed as described in the QWERTY task.

## OPTI-II task

The purpose of this task was to provide a potentially more efficient keyboard layout than the QWERTY or ABCDEF layouts for a continuous cursor communication interface. The conventional QWERTY layout is not ideal for selecting letters via continuous cursor navigation. Thus, a more efficient keyboard layout that minimizes the average distance travelled between letters should increase the typing rate. We used the OPTI-II keyboard layout described in the HCI literature (*Rick, 2010*) as an optimized layout for text entry with a continuous cursor. This was used with both participants, with timings and promptings identical to the QWERTY and ABCDEF tasks. For participant T6, a programming error caused the accessible workspace for the OPTI-II task (copy typing) to stop in the middle of the bottom row of keys (contrary to other tasks, where the accessible workspace extended past the keyboards).

## Metrics

The performance of each task was measured using one of two metrics, depending on the task. Performance on the grid task was measured via achieved bitrate, measured in bits per second, and performance on the typing tasks (QWERTY, ABCDEF, OPTI-II) was measured via correct characters per minute.

### Achieved bitrate

The grid task, representing a stable, memoryless, discrete communication channel with random, uniformly-distributed prompted targets, satisfies information theoretic criteria for measuring achieved bitrate (*Nuyujukian et al., 2015*). Achieved bitrate is a conservative measure of the actualized throughput of a communication channel. The achieved bitrate, B, is calculated via the following equation:

$$B = \frac{\log_2(N-1)\max(S-2E,0)}{t}$$

where *N* is the number of targets on the screen, *S* is the number of selections, *E* is the number of errors, and *t* is the time elapsed. The floor of this value is 0, since bitrate cannot be less than zero. Note that trials in which the participant timed out and made no selection are not counted in *S* or *E*, but are included in the value for time elapsed. This metric, in bits per second, represents the minimum expected throughput achievable from the system.

### Correct characters per minute

Typing rates were measured by calculating the number of correct characters transmitted over time (correct characters per minute [*Bacher et al., 2015*]). Correct characters were defined as those that were not subsequently deleted by the participant using the delete key. This measure, C, is defined by the following equation:

$$C = \frac{\max(S-2D, 0)}{t}$$

where *S* is the number of selections, *D* is the number of delete key selections, and *t* is the elapsed time.

We note that this metric labels typographical errors or spelling errors as correct characters. However, as it is not clear whether the participant was aware of a given spelling error, we only considered errors those that were actually deleted. This metric also parallels achieved bitrate, in that it only measures the net characters transmitted over time.

## Quantifying movement suppression

For sessions in which participant T6 was asked to suppress her movements to the best of her abilities (*Figure 4*), we first quantified the degree to which movements were suppressed during decoder calibration (*Figure 4—figure supplement 1*). As mentioned earlier, finger movements were measured using a dataglove (5DT, Irvine, CA). For each condition (i.e., freely moving vs. suppressed movement), we measured the finger position as a function of time (relative to the starting position for each trial), and averaged these positions across all trials for a given target direction. To robustly

evaluate the degree of suppression between freely moving and suppressed movement, we compared the time epochs spanning 600–1200 ms after target onset, which was well after movement was detectable but before movements became more variable across trials (i.e., to perform corrective movements as T6 approached the target). Movement suppression was only estimated for target directions in which movement on a given finger was to be expected to avoid singular values (e.g., as index finger movements were related to control of the horizontal dimension, index finger movements were not compared for the vertical targets where they would be expected to be 0). Data compared are from T6's trial days 570 (freely moving) and 595 (movements suppressed). We next quantified the degree to which movements were suppressed during closed-loop BCI control (Grid task). Individual trials were grouped by the target direction (i.e., the angle between the previous target and the prompted target for the current trial; eight possible directions) and finger positions were averaged across all trials of a given direction. Trials that lasted less than 1200 ms were excluded from the analysis. To ensure that any minute movements were captured, movements were quantified using the absolute value (rather than the signed value) of the finger position at each time point relative to the starting position for each trial. Analysis includes all Grid task data from T6's trial days 595 and 598 (movements suppressed). Unfortunately, for freely moving sessions, finger positions were not recorded during closed-loop BCI control, so data are unavailable for the specific comparison of finger movements during closed-loop BCI control for freely moving vs. movement suppressed sessions.

## Code availability

Code, which is platform specific and implemented in xPC, may be made available upon request to corresponding author.

## Acknowledgements

The authors would like to thank participants T6, T5, T7, and their families and caregivers; EN Eskandar for T7 implantation surgery; B Davis, B Pedrick, E Casteneda, M Coburn, S Patnaik, P Rezaii, B Travers, and D Rosler for administrative support; SI Ryu for surgical assistance; L Barefoot, S Cash, J Menon, and S Mernoff for clinical assistance; A Sarma, and N Schmansky for technical assistance; V Gilja, JD Simeral, JA Perge and B Jarosiewicz for technical assistance and helpful scientific discussions; JP Donoghue for helpful scientific discussions.

This work was supported by: Stanford Office of Postdoctoral Affairs; Craig H Neilsen Foundation; Stanford Medical Scientist Training Program; Stanford BioX-NeuroVentures, Stanford Institute for Neuro-Innovation and Translational Neuroscience; Larry and Pamela Garlick; Samuel and Betsy Reeves; NIH-NIDCD R01DC014034; NIH-NINDS R01NS066311; NIH-NIDCD R01DC009899; NIH-NICHD-NCMRR (N01HD53403 and N01HD10018); Rehabilitation Research and Development Service, Department of Veterans Affairs (B6453R); MGH-Deane Institute for Integrated Research on Atrial Fibrillation and Stroke; Executive Committee on Research, Massachusetts General Hospital.

The content is solely the responsibility of the authors and does not necessarily represent the official views of the National Institutes of Health, the Department of Veterans Affairs, or the United States Government. CAUTION: Investigational Device. Limited by Federal Law to Investigational Use.

## Additional information

### Funding

| Funder | Grant reference number | Author |
| --- | --- | --- |
| Craig H. Neilsen Foundation | Postdoctoral Fellowship | Chethan Pandarinath |
| Stanford Medical Scientist Training Program | | Paul Nuyujukian |
| U.S. Department of Veterans Affairs | B6453R | Leigh R Hochberg |
| Massachusetts General Hospi- | Deane Institute for | Leigh R Hochberg |

| | | |
|---|---|---|
| tal | | Integrated Research on Atrial Fibrillation and Stroke |
| National Institute on Deafness and Other Communication Disorders | R01DC009899 | Leigh R Hochberg |
| Eunice Kennedy Shriver National Institute of Child Health and Human Development | N01HD10018 | Leigh R Hochberg |
| Eunice Kennedy Shriver National Institute of Child Health and Human Development | N01HD53403 | Leigh R Hochberg |
| National Institute of Neurological Disorders and Stroke | R01NS066311 | Krishna V Shenoy Jaimie M Henderson |
| National Institute on Deafness and Other Communication Disorders | R01DC014034 | Krishna V Shenoy Jaimie M Henderson |
| Stanford University | BioX-NeuroVentures | Krishna V Shenoy Jaimie M Henderson |
| Stanford Institute for Neuro-Innovation and Translational Neuroscience | | Krishna V Shenoy Jaimie M Henderson |
| Larry and Pamela Garlick | | Jaimie M Henderson |
| Samuel and Betsy Reeves | | Jaimie M Henderson |

The funders had no role in study design, data collection and interpretation, or the decision to submit the work for publication.

## Author contributions

CP, PN, Responsible for study design, research infrastructure development, algorithm design, data collection, analysis, and manuscript preparation; CHB, Contributed to study design and data collection for participants T6 and T5; BLS, Contributed to study design and data collection from participant T7; JS, Contributed to technical development; FRW, Contributed to algorithm design; LRH, Contributed to study design and is the sponsor-investigator of the multi-site pilot clinical trial; KVS, Was involved in all aspects of the study; JMH, Was responsible for surgical implantation for study participants T6 and T5 and was involved in all aspects of the study

## Author ORCIDs

Chethan Pandarinath, http://orcid.org/0000-0003-1241-1432
Leigh R Hochberg, http://orcid.org/0000-0003-0261-2273
Jaimie M Henderson, http://orcid.org/0000-0002-3276-2267

## Ethics

Clinical trial registration NCT00912041

Human subjects: Permission for these studies was granted by the US Food and Drug Administration (Investigational Device Exemption) and Institutional Review Boards of Stanford University (protocol # 20804), Partners Healthcare/Massachusetts General Hospital (2011P001036), Providence VA Medical Center (2011-009), and Brown University (0809992560). The three participants in this study, T5, T6 and T7, were enrolled in a pilot clinical trial of the BrainGate Neural Interface System (http://www.clinicaltrials.gov/ct2/show/NCT00912041). Informed consent, including consent to publish, was obtained from the participants prior to their enrollment in the study.

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
