## [Decision Letter]

Thank you for submitting your article "High performance communication by people with ALS using an intracortical brain-computer interface" for consideration by *eLife*. Your article has been reviewed by two peer reviewers, and the evaluation has been overseen by Sabine Kastner as the Reviewing Editor and enior Editor. The following individual involved in review of your submission has agreed to reveal his identity: Hagai Lalazar (Reviewer #3).

The reviewers have discussed the reviews with one another and the Reviewing Editor has drafted this decision to help you prepare a revised submission.

Summary: This study examines spelling performance for two patients with ALS using an intracortical brain computer interface. This is the most comprehensive and elegant study that has been performed on this topic up to this point, and the editors agreed that it merits further consideration for publication at *eLife*.

Essential revisions:

1) Please discuss why the classification approach (vs. regression underlying cursor control) has not yet been explored.

2) Please provide some evidence that during the sessions when subject T6 was instructed not to use her hand, there was no overt movement. If you don't have EMG or motion capture, even an analysis of the video might satisfy this concern.

3) Please provide additional data (ideally EMG/high resolution eye tracking) to rule out muscle artifacts, or, alternatively, careful documentation of raw LFP traces that were used for control, demonstrating that they appear "naturalistic" rather than "artefactual".

4) Please explain why the HMM was not used with subject T7.

5) Please provide a detailed training history of the subjects (including previous experiments) and obtain as complete as possible training history for the studies they are comparing with. Please explain why the comparison is justified.

*Reviewer #1:*

Overview: The current study assesses spelling performance for two patients with ALS using an intracortical Brain computer interface (BCI) and a specific algorithm (refit-KF for cursor movement and an HMM for click selection). The main claim of the study is that the exhibited performance is better than previous spelling BCI studies with disabled patients.

General assessment: My main concern is whether the study's approach and result are innovative enough for publication in *eLife*. The authors previously published a study with the same participants (Gilja et al., 2015) using the same algorithm for cursor movement control. In the current study this is additionally combined with click selection (via an HMM) for spelling, which in the attached IEEE paper is shown to provide a modest but significant improvement on target selection. However, in one of the 2 subjects in the current study, click selection wasn't used. Thus, it is unclear how important is this addition for the exhibited performance. Relatedly, a recent noninvasive BCI spelling study on healthy subjects (Townsend & Platsko, 2016) appear to exhibit similar performance to the current study (1.73 bps in their "free spelling" task, see their table 5). That study should be referred to and the comparison discussed. While the current study appears to exhibit the best spelling accuracy to date as measured on ALS patients using a BCI, another study (McCane et al., 2015) showed that ALS patents and age matched controls exhibit similar performance while using the noninvasive BCI speller used in the Townsend & Platsko study. Thus, that study results arguably apply to ALS patients as well.

Other issues:

1) Intracranial LFP signals can be affected by activity of head muscles, including eye muscles. I'm concerned that such artifacts took place in the higher performing subject (T6), especially given that for that subject LFP frequencies for up to 450 Hz were used, as such high frequencies are more prone to artifacts than low frequencies. If possible, the authors should perform an additional experimental session where EMG signals from head muscles are recorded, or at least eye movements (including micro-saccades) are tracked, and show that the relevant control signals are uncorrelated with the EMG/eye movements.

2) Given that subject T6 has used an eye gaze speller before, it should be feasible to compare the spelling performance of that system vs. the one in the current study. This should be done to substantiate the main conclusion of the study that "intracortical BCIs offer a promising approach to assistive communication systems for people with paralysis."

3) Related to the above, and given that the paper is relatively short, there should be added to the Discussion a section detailing a cost-benefit analysis of invasive methods vs. noninvasive ones. If the authors believe (as I assume they do) that the exhibited performance of their BCI outweighs the risks of invasive surgery and chronic implants then that claim should be made explicit and substantiated.

4) The number of subjects (2) is standard for this type of studies. However, there are significant differences in the procedures involving each of the subjects, particularly regarding: 1 – the calibration procedure of the algorithm, which in one subject (T6) involved performing motor movements with the fingers. 2 – one participant (T6) performed clicks which were decoded using an HMM, the other (T7) did not. Thus, it is unclear if the subjects could actually be pooled together.

5) Related to the above, the best subject in the study seemed to exhibit a capacity for natural motor control not attained by the other subject in the study, and possibly by some subjects in previous studies that are being compared. Thus, this capacity may underlie the exhibited high performance. In order to address this concern the authors performed an additional two sessions with this subject were the movements were required to be suppressed. However, it is unclear if movements were actually suppressed in those sessions, as no data to that effect (such as EMG) is shown. Thus, it is unclear if this control is adequate. The authors should substantiate the claim of this control by showing such data.

6) I couldn't find references for several of the studies that are being compared in the table (Figure 3 – —figure supplement 3).

7) The best performing subject (T6) barely has any spiking signal in the array (Figure 3 – —figure supplement 5.) Thus, it is unclear what type of activity is being extracted and used. The authors should provide traces of the threshold crossings from a small number of channels (e.g. 10) which were most used for control, for example as assessed by off-line, channel dropping analysis. Additionally they should provide traces of the LFP features used for control, assessed in the same way. There should also be an analysis detailing which features were more influential for control (spiking or LFP) for both cursor movement and click selection.

8) There should be a summarized but detailed history of BCI experience for both subjects as this is directly relevant to their capacity to control the BCI. Differences in training can make it hard to compare between studies. Subjects in intracortical BCI studies typically train for much longer periods than in EEG BCI studies. For example, in (Mainsah et al., 2015) which is a study that should be added to the comparison table, subjects trained for 3 days. While subjects in the current study may have well been training for months in different BCI experiments. The authors should (when possible) add that information to the study comparison table and explain, in cases where previous studies had much shorter training periods than the current study, why the comparison is still justified.

References:

Townsend, G. & Platsko, V. Pushing the P300-based brain-computer interface beyond 100 bpm: extending performance guided constraints into the temporal domain. J Neural Eng., 13(2):026024. (2016).

McCane, L.M., Heckman, S.M., McFarland, D.J., Townsend. G., Mak, J.N., Sellers, E.W., Zeitlin, D., et al. P300-based brain-computer interface (BCI) event-related potentials (ERPs): People with amyotrophic lateral sclerosis (ALS) vs. age-matched controls. Clinical neurophysiology 126 (11): 2124-31. (2015).

Mainsah, B.O., Collins, L.M., Colwell, K.A., Sellers, E.W., Ryan, D.B., Caves, K. & Throckmorton, C.S. Increasing BCI communication rates with dynamic stopping towards more practical use: an ALS study. J Neural Eng., 12(1): 016013. (2015).

*Reviewer #3:*

A closed-loop BMI for typing was tested with 2 ALS patients. Performance showed substantial improvement compared to previous studies, on both realistic and quantitative tests. This study is thorough, its details are, for the most part, explained well, and has important clinical implications. I have made suggestions to improve the clarity and presentation of the manuscript.

1) Improving typing speeds can be evaluated both on an absolute scale (relative to the typing speeds of able-bodied typers), as well as, the improvement relative to previous studies. This study makes a substantial improvement in both regards, however both points can be made more clearly and earlier in the paper.

The comparison to able-bodied typers (main text, twelfth paragraph) should be stated in the Abstract (perhaps just comparing to texting, which is the most comparable to cursor control) and mentioned earlier in the manuscript (e.g. after the fourth paragraph of the main text). This is critical for a non-expert reader to be able to assess this work in the general context of the state-of-the-art of assistive communication devices.

Additionally, the table in Figure 3—figure supplement 3, should be made a full figure. This comparison gives a broad picture of the improved performance achieved in the current study (i.e. the variance across many other studies and the 1-2 orders of magnitude improvement over EEG based approaches). This table should have an additional column, describing in a few words the algorithm and type of user-interface used in each study (e.g. "p300-speller", "ReFIT-KF & HMM", etc.). Also, all the papers cited in the table should appear in the references.

2) Typing is inherently a (multi-class) classification problem, and not the regression problem underlying 2D cursor control. BCI typing speeds will eventually be limited only by the mean time of each cursor trajectory (between each letter selected), and any time used for the selection itself (dwell times, or "click"). This difference explains why the texting speeds of able-bodied subjects (which usually use only one finger) are lower that their keyboard typing speeds (which have discretized the alphabet into the 10 fingers). This issue has not been explored in the current paper, and to my knowledge, in the series of studies on "typing" with humans or monkeys (except for Andersen, et al. (2004)). If the authors have done any offline data-analysis to test this idea, it would be very helpful to describe it briefly. Otherwise, why this approach has not yet been explored should be explained.

3) Figure 3—figure supplement 1, discusses that performance (in both the copy-typing and grid tasks) was not significantly different when the subject was asked to suppress any overt movements. However, there is no measurement of the movement and its ensuing reduction after the instruction. The main text should be more forthcoming about this, and mention that this is only what the subject was trying to do, however was not measured and quantified. Moreover, as EMG from the arm was not measured, it should be mentioned that an effect of nascent muscle commands (that did not elicit observable movements) on the performance cannot be ruled out. This is especially important, as subject T6 had better results, and her remaining finger movements were used to train her initial decoders. As different patients may or may not have specific residual movements, this makes performance comparisons between them less precise. This point should also be mentioned, in the paragraph hypothesizing about the reasons for the performance differences between the subjects (main text, fifteenth paragraph).

4) The analysis in Figure 3—figure supplement 4 suffers from all the weaknesses of analyzing the tuning of M1 neurons by fitting them to a cosine-tuning function for movement direction, reported in many studies (some even by authors of the current study). For example, (i) the percent of neurons that show a change in preferred direction depends on the goodness-of-fit of the cosine model across the population (which the authors don't report), (ii) preferred directions have been shown to change as a function of time during the movement (Churchland & Shenoy (2007), Figure 13), (iii) there are high frequency deviations from cosine tuning, which, in addition to the cosine tuning component, may be expected from random connectivity (Lalazar, Abbott, Vaadia (2016)), etc. This figure and the associated sentence in the main text (main text, eighth paragraph; and Methods subsection) do not contribute to the manuscript and only diminish from its otherwise compelling level of rigor. I suggest removing them.

5) Why was the HMM decoder not used for subject T7? This should be explained.

[Editors' note: further revisions were requested prior to acceptance, as described below.]

Thank you for resubmitting your work entitled "High performance communication by people with paralysis using an intracortical brain-computer interface" for further consideration at *eLife*. Your revised article has been favorably evaluated by Sabine Kastner (Senior Editor) and one of the previous reviewers.

The manuscript has been improved but there are some remaining issues that need to be addressed before acceptance, as outlined below:

1) The paragraph discussing the costs and benefits of invasive vs. noninvasive bci strategies should be more balanced. The potential risk of brain surgery (e.g. infection, tissue damage, brain swelling, seizures) needs to be explicitly stated, especially the risk of infection given that there is an implant providing a physical connection from the brain to outside the scalp.

2) – On several instances, the authors stress the benefits of a fully self-calibrating, fully wireless implantable system. They need to make clear that theirs is not such a system.

---

## [Author Response]

*Essential revisions:*

*1) Please discuss why the classification approach (vs. regression underlying cursor control) has not yet been explored.*

We agree that classification (i.e., estimating the endpoint goal) is a very promising alternative approach for use in communication interfaces. We have added the text included below to the Discussion section of the main manuscript in order to clarify why the current approach was investigated:

“Previous work with non-human primates from our lab and others (e.g., Shenoy et al., NeuroReport2003; Musallam et al., Science2004; Santhanam et al., Nature2006) demonstrated that BCI strategies which leverage discrete classification can achieve high communication rates. […] Thus, there are multiple technical and scientific challenges to address, and developing these approaches for clinical trial participants is an active area of research.”

*2) Please provide some evidence that during the sessions when subject T6 was instructed not to use her hand, there was no overt movement. If you don't have EMG or motion capture, even an analysis of the video might satisfy this concern.*

This request refers to sessions with participant T6, who at the time of this study still retained the ability to make dexterous movements with her fingers. A potential concern was that T6’s high performance might be due to her ability to make movements. To address this, in the original submission, we performed control experiments in which the participant was asked to suppress her natural movements as best as possible and control the BCI. In these sessions, decoders were calibrated with physical movements suppressed (i.e., “open-loop” calibration), and the participant imagined movements of her fingers. These experiments showed that performance in closed-loop BCI control with movements suppressed was comparable to performance when we did not explicitly ask her to suppress hand movements.

Further, to address questions regarding the original data on movement suppression with participant T6, we have added quantification to show the degree to which T6’s movements were suppressed. This quantification is based on hand movement data from a commercially-available “data glove” sensor system, which was used to track the position of the participant’s fingers during the session. This analysis is now presented in detail in Figure 4 and its supplements.

Overall, when T6 actively attempted to suppress movements, her movement was reduced by a factor of 7.2 – 12.6 (Figure 4—figure supplement 1). As described in the main text, despite this factor of 7.2 – 12.6 in movement suppression, performance was quite similar to performance when T6 moved freely. Across all three quantitative evaluation types (OPTI-II, QWERTY & Grid), the performance differences were within 0-20% and not significant (p > 0.2 in all cases, Student’s *t* test).

The concern that movement ability is required to achieve high-performance BCI control is addressed by data from a new participant (T5). Unlike T6, T5 had no ability to make any functional movements of his upper and lower extremities, and retained only very limited volitional movements of his non-dominant elbow and fingers (detailed in Methods). This level of movement ability is representative of the severely motor impaired population. For T5, continuous cursor control was based on attempted movement of the dominant hand/arm (for which no consistent movements were possible), and discrete selection was based on attempted movement of the non-dominant hand. Despite T5’s large decrease in movement ability relative to participant T6, his performance was in fact higher than participant T6. This further supports the idea that the ability to generate movements is not required for high performance cursor control.

As requested by the reviewers, we have adjusted the manuscript to make the comparison between T6’s performance with or without suppressing movements more prominent. We moved the text describing this topic into its own paragraph in the Results section, we moved the comparison to its own dedicated figure with two new supplementary figures (Figure 4 and Figure 4—figure supplement 1 and 2), and we have added text to the Results and Discussion to highlight this issue and detail our conclusions.

Results:

“As might be expected, T6 found that suppressing her natural movement was a challenging, cognitively demanding task. […] Despite this factor of 7.2 – 12.6 in movement suppression, performance was quite similar to performance when T6 moved freely – across all three quantitative evaluation types (Grid, OPTI-II, QWERTY), the performance differences were within 0-20% and not significant (p > 0.2 in all cases, Student’s *t* test).”

Discussion paragraphs:

“At the time of this study, participant T6 still retained the ability to make dexterous movements of her hands and fingers, which may raise the question of whether her high level of performance was related to the generation of movement. […] Further, there was little if any correspondence between the participants’ movement abilities and BCI performance.”

*3) Please provide additional data (ideally EMG/high resolution eye tracking) to rule out muscle artifacts, or, alternatively, careful documentation of raw LFP traces that were used for control, demonstrating that they appear "naturalistic" rather than "artefactual".*

We appreciate the reviewer’s question regarding high-frequency LFP signals (which were used for BCI control with participant T6) and whether these might contain artifacts related to EMG from eye movements. As requested, we have provided both additional data and analyses to rule out this possibility. We have added the following text to the Methods that discusses the possibility and our new data/analyses in detail:

“A potential concern with decoding a power signal such as these high frequency LFP (HF-LFP) signals (which were used for participant T6) is that they may pick up artifacts related to EMG from eye movements. […] These lines of evidence make the possibility that HF-LFP signals are eye movement-related highly unlikely.”

Last, as a piece of anecdotal evidence, we note that we originally aimed to conduct these performance evaluations with T6 seated at a closer distance to the display (56 cm). However, at this distance, we found that T6 had to make large eye movements to scan the workspace in order to find letters and targets, which hampered her performance and comfort. Instead, we performed the evaluations at a farther distance from the display (90 cm), greatly reducing her eye movements and improving her performance and comfort. If eye movement-related signals were mediating performance, we would expect performance to suffer as eye movements were reduced. However, this was not the case.

*4) Please explain why the HMM was not used with subject T7.*

Both reviewers brought up the issue that the HMM was not replicated with participant T7, and we understand and appreciate this question. We have added text to the Discussion (replicated below) that explains why this was not possible. Further, to directly address this question, in the revised submission we added data from a third participant (T5), which allowed further replication of our methods, including the HMM. Overall performance with T5 was even higher than the two previous participants, and further, the HMM was even more accurate with T5 than with T6 (as documented in Figure 3—figure supplement 1).

The following was added to the Discussion to explain the lack of HMM with participant T7:

“Both participants T6 and T5 used the HMM decoder for discrete selection. Our goal was to also use the HMM with participant T7. […] Unfortunately, however, T7 passed away before the HMM sessions could be conducted.”

*5) Please provide a detailed training history of the subjects (including previous experiments) and obtain as complete as possible training history for the studies they are comparing with. Please explain why the comparison is justified.*

We appreciate the reviewer’s comments regarding training history for our study and the studies we compare to in Table 1. To address these comments, we added a table to summarize the training history as best we can for all studies in Table 1 (new Table 1).

At the time of the sessions presented in this paper, both T6 & T7 had performed 2-3 research sessions per week for 1.5 years. Sessions consisted mainly of tasks related to controlling computer cursors, typing and communication, and attempted / imagined movement imagery, with additional sessions dedicated to robotic arm control. We did not perform detailed experiments to track learning or the effects of BCI experience on performance. Fortunately, for T6 & T7, a direct comparison is possible with Jarosiewicz et al., Science Translational Medicine2015 (which, prior to this study, had the highest reported bitrates for a BCI used by a person with motor impairment) as both T6 & T7 were participants in that study as well. For T7, later sessions from Jarosiewicz et al. were performed within 3 months of the current study. For T6, the sessions from Jarosiewicz et al. were collected both before and after the current sessions. For both participants, as noted in the text, performance in the current study was approximately a factor of 2 higher than Jarosiewicz et al. This rules out training time as a main driver of the differences in performance.

Further, we have now added data from a new, recently-implanted participant (T5; implanted in Aug 2016, first BCI session Sept 2016, and data collected in Oct 2016) which shows that higher performance than T6 could be achieved within 9 sessions of first controlling the BCI. We were not trying to minimize this time in any way, and spent many of those first sessions simply characterizing the types of intended movements that elicit neural modulation in the areas recorded by the arrays.

While it may be possible, as the reviewer suggests, that EEG studies may increase performance with longer exposure to the BCI, this has yet to be shown. As mentioned in Table 1, the Sellers and colleagues paper tracked an EEG participant for > 1 year, and they did not see an increase in performance over this time. Similarly, the average participant in the Mugler and colleagues study had 3+ years of experience with BCIs, yet performance was not markedly higher than the other P300 EEG-based studies in the literature.

As mentioned above, we did not perform detailed experiments to track learning. In our experience, large changes in performance were achieved by trying out new algorithms or behavioral imagery paradigms, but we did not generally see slow gradual increases in BCI control quality from repeatedly using the same decoding strategies. A recent non-human primate report from our group is consistent with the idea of seeing little if any change in performance across months and years of BCI use (Gilja et al., Nature Neuroscience2012, see Figure 2) when the decoder algorithm provides high performance early on and thus there is little “pressure” on the system to engage neural adaptation (Shenoy & Carmena, Neuron, 2014).

*Reviewer #1:*

*Overview: The current study assess spelling performance for two patients with ALS using an intracortical Brain computer interface (BCI) and a specific algorithm (refit-KF for cursor movement and an HMM for click selection). The main claim of the study is that the exhibited performance is better than previous spelling BCI studies with disabled patients.*

*General assessment: My main concern is whether the study's approach and result are innovative enough for publication in eLife. The authors previously published a study with the same participants (Gilja et al., 2015) using the same algorithm for cursor movement control. In the current study this is additionally combined with click selection (via an HMM) for spelling, which in the attached IEEE paper is shown to provide a modest but significant improvement on target selection. However, in one of the 2 subjects in the current study, click selection wasn't used. Thus, it is unclear how important is this addition for the exhibited performance.*

Thank you for this helpful question. We agree that this work builds on previous methodologies, but there are several reasons why the current work represents an important milestone in the development of clinically useful BCIs. While our previous manuscript (Gilja et al., 2015) showed improved continuous control performance over earlier approaches, as the reviewer points out, the current study adds a parallel decoding method to enable discrete selection (the HMM), which is a critical step in developing point-and-click interfaces that would be suitable to control a general purpose computing device. While it is unfortunate that the HMM could not be tested with the participant T7 (as described in Essential revisions #4 above), we now replicate the HMM results with an additional participant (T5) and show even higher performance than in the original manuscript.

Importantly, following the Gilja et al. study, it was unknown whether high performance BCI control based on motor cortical signals could be maintained in the face of increasingly complex tasks and cognitive loads. Here we explicitly show this. In all tasks presented, the participants had to select the correct targets without accidentally selecting distractor targets. Further, in the copy typing tasks here, the participants were required to copy sequences of words, keep track of their position within the word and sentence, recall their intended letter’s position (across multiple keyboard layouts), and navigate to the correct letter. In the free typing task, the participant had to actively construct their intended sentences while controlling the BCI. In all cases, the system demonstrated unparalleled performance for a BCI with subjects with motor impairment.

Further, this study documents a communication BCI with people with motor impairment whose performance would satisfy the majority of users’ desires. Specifically, as mentioned in the text, a previous survey of the communication desires of people with ALS (Huggins et al., Amyotroph Lateral Scler2011) found that 59% of respondents would be satisfied with a communication speed of 10-14 characters per minute (and 72% with 15-19 characters per minute). The typing performance reported in the current study (31.6 ccpm, 39.2 ccpm (7.8 wpm), and 13.5 ccpm for T6, T5, and T7, respectively) would, at least as documented by Huggins’ study, satisfy the desires of the majority of this population (and this performance would further increase with the addition of common word completion algorithms).

*Relatedly, a recent noninvasive BCI spelling study on healthy subjects (Townsend & Platsko, 2016) appear to exhibit similar performance to the current study (1.73 bps in their "free spelling" task, see their table 5). That study should be referred to and the comparison discussed. While the current study appears to exhibit the best spelling accuracy to date as measured on ALS patients using a BCI, another study (McCane et al., 2015) showed that ALS patents and age matched controls exhibit similar performance while using the noninvasive BCI speller used in the Townsend & Platsko study. Thus, that study results arguably applies to ALS patients as well.*

Thank you for raising the Townsend and Platsko work. That study and a few other previous BCI studies (e.g., Brunner et al., Front Neurosci2011, Spuler et al., PLoS ONE2012, Chen et al., PNAS2015) nominally report higher typing rates than those listed in the table when measured with healthy subjects. However, we exclude these from the comparison as, at present, we do not know how their reported performance would translate to subjects with motor impairment.

*Other issues:*

*1) Intracranial LFP signals can be affected by activity of head muscles, including eye muscles. I'm concerned that such artifacts took place in the higher performing subject (T6), especially given that for that subject LFP frequencies for up to 450 Hz were used, as such high frequencies are more prone to artifacts than low frequencies. If possible, the authors should perform an additional experimental session where EMG signals from head muscles are recorded, or at least eye movements (including micro-saccades) are tracked, and show that the relevant control signals are uncorrelated with the EMG/eye movements.*

We appreciate this question. The detailed response to this question is in Essential revisions #3 above.

*2) Given that subject T6 has used an eye gaze speller before, it should be feasible to compare the spelling performance of that system vs. the one in the current study. This should be done to substantiate the main conclusion of the study that "intracortical BCIs offer a promising approach to assistive communication systems for people with paralysis."*

We appreciate this important question. First, participant T6 is no longer a part of the clinical trial, therefore collecting such data is not possible. Second, we hold a slightly different perspective than that posed in this reviewer question. We believe that our results presented support the conclusion that intracortical BCIs offer a promising approach to assistive communication systems for people with paralysis. The results show that intracortical BCIs are capable of high typing rates (12-40 correct characters per minute (ccpm)) and communication rates (1.4-3.7 bits per second), and were demonstrated in complex, real-world tasks. As mentioned in the paper and above (B1), the levels of communication performance demonstrated in our study would satisfy the majority of people with ALS. We believe that this is sufficient to conclude that intracortical BCIs offer a promising approach to assistive communication systems for people with paralysis.

In a separate study, we are actively directly comparing available AAC systems to the current decoders used with our iBCI technology. This turns out to be a much more difficult study to perform than one would imagine (due largely to the many ways in which eye gaze and other AAC systems fail and require recalibration, and due to challenges in assessing the benefits provided by the “input” (eye gaze vs. neural control) as compared to the interface (screen keyboard differences, dwell time vs. click selection, etc.).

*3) Related to the above, and given that the paper is relatively short, there should be added to the Discussion a section detailing a cost-benefit analysis of invasive methods vs. noninvasive ones. If the authors believe (as I assume they do) that the exhibited performance of their BCI outweighs the risks of invasive surgery and chronic implants then that claim should be made explicit and substantiated.*

As requested, we have added the following to the Discussion section:

“The question of the suitability of implanted versus external BCI systems (or any other external AAC system) for restoring function is an important one. Any technology (or any medical procedure) that requires surgery will be accompanied by some risk. […] Thus, there is a clear willingness among people with paralysis to undergo a surgical procedure if it could provide significant improvements in their daily functioning.”

*4) The number of subjects (2) is standard for this type of studies. However, there are significant differences in the procedures involving each of the subjects, particularly regarding: 1 – the calibration procedure of the algorithm, which in one subject (T6) involved performing motor movements with the fingers. 2 – one participant (T6) performed clicks which were decoded using an HMM, the other (T7) did not. Thus, it is unclear if the subjects could actually be pooled together.*

We appreciate this question. The detailed response is in Essential revisions #4 above, including the addition of a new participant (T5) that addresses quite directly these questions.

*5) Related to the above, the best subject in the study seemed to exhibit a capacity for natural motor control not attained by the other subject in the study, and possibly by some subjects in previous studies that are being compared. Thus, this capacity may underlie the exhibited high performance. In order to address this concern the authors performed an additional two sessions with this subject were the movements were required to be suppressed. However, it is unclear if movements were actually suppressed in those sessions, as no data to that effect (such as EMG) is shown. Thus, it is unclear if this control is adequate. The authors should substantiate the claim of this control by showing such data.*

We appreciate this question. The detailed response is in Essential revisions #2 above.

*6) I couldn't find references for several of the studies that are being compared in the table (Figure 3 – —figure supplement 3).*

This is now fixed. We apologize for this error in referencing.

*7) The best performing subject (T6) barely has any spiking signal in the array (Figure 3 – —figure supplement 5.) Thus, it is unclear what type of activity is being extracted and used. The authors should provide traces of the threshold crossings from a small number of channels (e.g. 10) which were most used for control, for example as assessed by off-line, channel dropping analysis. Additionally they should provide traces of the LFP features used for control, assessed in the same way. There should also be an analysis detailing which features were more influential for control (spiking or LFP) for both cursor movement and click selection.*

We appreciate this question. The detailed response is in Essential revisions #3 above.

*8) There should be a summarized but detailed history of BCI experience for both subjects as this is directly relevant to their capacity to control the BCI. Differences in training can make it hard to compare between studies. Subjects in intracortical BCI studies typically train for much longer periods than in EEG BCI studies. For example, in (Mainsah et al., 2015) which is a study that should be added to the comparison table, subjects trained for 3 days. While subjects in the current study may have well been training for months in different BCI experiments. The authors should (when possible) add that information to the study comparison table and explain, in cases where previous studies had much shorter training periods than the current study, why the comparison is still justified.*

We appreciate this question. The detailed response is in Essential revisions #5 above. Also, we appreciate the additional reference and have now added this study to the comparison table.

*Reviewer #3:*

*A closed-loop BMI for typing was tested with 2 ALS patients. Performance showed substantial improvement compared to previous studies, on both realistic and quantitative tests. This study is thorough, its details are, for the most part, explained well, and has important clinical implications. I have made suggestions to improve the clarity and presentation of the manuscript.*

*1) Improving typing speeds can be evaluated both on an absolute scale (relative to the typing speeds of able-bodied typers), as well as, the improvement relative to previous studies. This study makes a substantial improvement in both regards, however both points can be made more clearly and earlier in the paper.*

*The comparison to able-bodied typers (main text, twelfth paragraph) should be stated in the Abstract (perhaps just comparing to texting, which is the most comparable to cursor control) and mentioned earlier in the manuscript (e.g. after the fourth paragraph of the main text). This is critical for a non-expert reader to be able to assess this work in the general context of the state-of-the-art of assistive communication devices.*

We appreciate this suggestion. However, we purposefully decided not to discuss absolute performance numbers (i.e., raw typing speed or communication rates) in the Abstract because our goal in the Abstract is to put this work in context in comparison to previous work, in order to highlight the advance of this paper. We chose instead to reserve the raw performance metrics for the results. We believe that the most appropriate place to relate these performance metrics to broader communication methods (e.g., typing by healthy subjects) is the Discussion, and have now moved the text relating to communication rates for healthy subjects to the first paragraph of the Discussion. In sum, we can certainly appreciate how this suggestion makes sense and could help many appreciate this work even more, but in balance we believe that the present approach has some key advantages.

*Additionally, the table in Figure 3—figure supplement 3, should be made a full figure. This comparison gives a broad picture of the improved performance achieved in the current study (i.e. the variance across many other studies and the 1-2 orders of magnitude improvement over EEG based approaches).*

We agree. This is now Table 1.

*This table should have an additional column, describing in a few words the algorithm and type of user-interface used in each study (e.g. "p300-speller", "ReFIT-KF & HMM", etc.).*

This has been added to the table. Thank you for the suggestion.

*Also, all the papers cited in the table should appear in the references.*

We apologize for this error in referencing. This is now fixed.

*2) Typing is inherently a (multi-class) classification problem, and not the regression problem underlying 2D cursor control. BCI typing speeds will eventually be limited only by the mean time of each cursor trajectory (between each letter selected), and any time used for the selection itself (dwell times, or "click"). This difference explains why the texting speeds of able-bodied subjects (which usually use only one finger) are lower that their keyboard typing speeds (which have discretized the alphabet into the 10 fingers). This issue has not been explored in the current paper, and to my knowledge, in the series of studies on "typing" with humans or monkeys (except for Andersen, et al. (2004)). If the authors have done any offline data-analysis to test this idea, it would be very helpful to describe it briefly. Otherwise, why this approach has not yet been explored should be explained.*

We appreciate this suggestion. The detailed response is in Essential revisions #1 above. Our response includes how we are quite familiar with this approach (so we doubly appreciate / enjoy receiving this question) as one of the senior authors (Shenoy) co-developed this approach with Richard Andersen years ago (Shenoy, …, Andersen NeuroReport2003) and, to our knowledge, has achieve the highest level of performance with the classification approach to date (Santhanam, …, Shenoy Nature2006). In future work we anticipate investigating this approach in people, as Prof. Andersen has beautifully been doing, but there are many advantages – importantly including flexibility with various interfaces – with the regression approach (as described above).

*3) Figure 3—figure supplement 1, discusses that performance (in both the copy-typing and grid tasks) was not significantly different when the subject was asked to suppress any overt movements. However, there is no measurement of the movement and its ensuing reduction after the instruction. The main text should be more forthcoming about this, and mention that this is only what the subject was trying to do, however was not measured and quantified. Moreover, as EMG from the arm was not measured, it should be mentioned that an effect of nascent muscle commands (that did not elicit observable movements) on the performance cannot be ruled out. This is especially important, as subject T6 had better results, and her remaining finger movements were used to train her initial decoders. As different patients may or may not have specific residual movements, this makes performance comparisons between them less precise. This point should also be mentioned, in the paragraph hypothesizing about the reasons for the performance differences between the subjects (main text, fifteenth paragraph).*

We appreciate this question and the suggestions. The detailed response is in Essential revisions #2 above.

An additional point to address this question: as mentioned in the section regarding “Performance of the BCI with movements suppressed” (Figure 4), we did not use finger movements to calibrate initial decoders in the “movement suppressed” sessions with T6. Rather, these decoders were calibrated in an “open-loop” fashion as the participant imagined movements. The degree of movement during this calibration is now detailed in Figure 4.

Last, regarding the impact of residual movements on performance – the addition of data from participant T5, who has no remaining functional movement but higher performance than either T6 or T7 – calls into question any potential relationship between movement ability and performance. Thus, we do not mention residual movement as a possible mediator of high performance, as the data do not support this conclusion. We have done our best to make the movement ability of each participant more clearly stated throughout the manuscript.

*4) The analysis in Figure 3—figure supplement 4 suffers from all the weaknesses of analyzing the tuning of M1 neurons by fitting them to a cosine-tuning function for movement direction, reported in many studies (some even by authors of the current study). For example, (i) the percent of neurons that show a change in preferred direction depends on the goodness-of-fit of the cosine model across the population (which the authors don't report), (ii) preferred directions have been shown to change as a function of time during the movement (Churchland & Shenoy (2007), Figure 13), (iii) there are high frequency deviations from cosine tuning, which, in addition to the cosine tuning component, may be expected from random connectivity (Lalazar, Abbott, Vaadia (2016)), etc. This figure and the associated sentence in the main text (main text, eighth paragraph; and Methods subsection) do not contribute to the manuscript and only diminish from its otherwise compelling level of rigor. I suggest removing them.*

We agree with the reviewer, and we are happy to remove it. This analysis (cosine-tuning across the neural population) has been removed. We included it originally as we are sometimes asked about this point, but we fully agree with the reviewer that this is not as meaningful an analysis as some might initially think and we surely wish to achieve and maintain a high level of rigor.

*5) Why was the HMM decoder not used for subject T7? This should be explained.*

This is detailed in Essential revisions #4 above.

[Editors' note: further revisions were requested prior to acceptance, as described below.]

*The manuscript has been improved but there are some remaining issues that need to be addressed before acceptance, as outlined below:*

*1) The paragraph discussing the costs and benefits of invasive vs. noninvasive bci strategies should be more balanced. The potential risk of brain surgery (e.g. infection, tissue damage, brain swelling, seizures) needs to be explicitly stated, especially the risk of infection given that there is an implant providing a physical connection from the brain to outside the scalp.*

We appreciate the desire to make the paragraph appropriately balanced, and have modified the text to list some of the immediate risks associated with the specific neurosurgical procedure. The paragraph already contains multiple references to guide readers to an in-depth discussion on the risks and risk-benefit analysis of an implanted neural interface.

“The question of the suitability of implanted versus external BCI systems (or any other external AAC system) for restoring function is an important one. […] That risk is not viewed in isolation, but is compared – by the individual contemplating the procedure – to the potential benefit [44,45]. […] Additional discussion of these topics are found in refs. [47,48].”

*2) On several instances, the authors stress the benefits of a fully self-calibrating, fully wireless implantable system. They need to make clear that theirs is not such a system.*

Thank you for this comment. There are two points in the paper that mention the prospect of a wireless and/or self-calibrating interface. We have modified them as below to prevent any confusion.

“A future self-calibrating, fully implanted wireless system could in principle be used without caregiver assistance, would have no cosmetic impact, and could be used around the clock. Such a system may be achievable by combining the advances in this report with previous advances in self-calibration and in fully-implantable wireless interfaces [15, 46].”

“In a recent survey of people with spinal cord injury [49], respondents with high cervical spinal cord injury would be more likely to adopt a hypothetical wireless intracortical system compared to an EEG cap with wires, by a margin of 52% to 39%.”

References cited above:

15) Jarosiewicz, B., Sarma, A. A., Bacher, D., Masse, N. Y., Simeral, J. D., Sorice, B., Oakley, E. M., Blabe, C., Pandarinath, C., Gilja, V., Cash, S. S., Eskandar, E. N., Friehs, G., Henderson, J. M., Shenoy, K. V., Donoghue, J. P., & Hochberg, L. R. Virtual typing by people with tetraplegia using a stabilized, self-calibrating intracortical brain-computer interface. Science Translational Medicine, 7, 313ra179. (2015)

44) Hochberg, L. & Cochrane, T. Implanted neural interfaces. Neuroethics Pract, 235, (2013)

45) Hochberg, L. R. & Anderson, K. D. BCI users and their needs. in Oxford University Press, (2012)

46) Borton, D. A., Yin, M., Aceros, J., & Nurmikko, A. An implantable wireless neural interface for recording cortical circuit dynamics in moving primates. J Neural Eng, 10, 026010. 10.1088/1741-2560/10/2/026010. (2013)

47) Ryu, S. I. & Shenoy, K. V. Human cortical prostheses: lost in translation? Neurosurgical focus, 27, E5. (2009)

48) Gilja, V., Chestek, C. A., Diester, I., Henderson, J. M., Deisseroth, K., & Shenoy, K. V. Challenges and opportunities for next-generation intracortically based neural prostheses. IEEE Transactions on Biomedical Engineering, 58, 1891--1899. (2011)

49) Blabe, C. H., Gilja, V., Chestek, C. A., Shenoy, K. V., Anderson, K. D., & Henderson, J. M. Assessment of brain--machine interfaces from the perspective of people with paralysis. Journal of neural engineering, 12, 043002. (2015)